# Identifying Policy Gradient Subspaces

**Jan Schneider**[1][*]     **Pierre Schumacher**[1,2]     **Simon Guist**[1]     **Le Chen**[1]
**Daniel Häufle**[2,3]     **Bernhard Schölkopf**[1]     **Dieter Büchler**[1]

[1]Max Planck Institute for Intelligent Systems, Tübingen, Germany
[2]Hertie Institute for Clinical Brain Research, Tübingen, Germany
[3]Institute for Computer Engineering, University of Heidelberg, Germany

## Abstract

Policy gradient methods hold great potential for solving complex continuous control tasks. Still, their training efficiency can be improved by exploiting structure within the optimization problem. Recent work indicates that supervised learning can be accelerated by leveraging the fact that gradients lie in a low-dimensional and slowly-changing subspace. In this paper, we conduct a thorough evaluation of this phenomenon for two popular deep policy gradient methods on various simulated benchmark tasks. Our results demonstrate the existence of such *gradient subspaces* despite the continuously changing data distribution inherent to reinforcement learning. These findings reveal promising directions for future work on more efficient reinforcement learning, e.g., through improving parameter-space exploration or enabling second-order optimization.

## 1 Introduction

Deep reinforcement learning (RL) has marked significant achievements in numerous challenging problems, ranging from Atari games (Mnih et al., 2013) to various real robotic challenges, such as contact-rich manipulation (Gu et al., 2017; Kalashnikov et al., 2018), complex planning problems (Everett et al., 2018; Ao et al., 2022), and hard-to-control dynamic tasks (Cheng et al., 2023; Kaufmann et al., 2023). Despite these notable successes, deep RL methods are often brittle due to the use of function approximators with large numbers of parameters and persistently changing data distributions – a setting notoriously hard for optimization. Deep RL, in its vanilla form, operates under limited prior knowledge and structural information about the problem, consequently requiring large numbers of interactions with the environment to reach good performance.

For supervised learning (SL), Gur-Ari et al. (2018) demonstrated that the gradients utilized for neural network optimization reside in a low-dimensional, slowly-changing subspace. Based on this insight, recent works introduce more structured optimization procedures for SL by identifying and harnessing these *gradient subspaces*. Exploiting this structure enables the optimization to be carried out in a reduced-dimensional subspace, yielding enhanced efficiency with minimal, if any, loss in performance (Li et al., 2018; Gressmann et al., 2020; Larsen et al., 2021; Li et al., 2022a).

Despite the benefits of subspace methods in SL, their adoption in deep RL has remained limited. A straightforward way to transfer these principles is to find lower-dimensional subspaces in policy gradient approaches (Peters & Schaal, 2008). Policy gradient (PG) methods estimate the gradient of the RL objective $\nabla_\theta J(\theta)$ to update the policy's parameters $\theta$ using some form of stochastic gradient descent (SGD). Since most SL approaches using subspaces operate at the level of the SGD optimization, PG algorithms would be a natural choice to leverage the knowledge about subspaces from SL in the RL context. Nevertheless, in RL, such methods have been explored primarily within the realm of evolutionary strategies (Maheswaranathan et al., 2019), representation learning (Le Lan et al., 2023), and transfer learning (Gaya et al., 2022). A possible explanation is the constantly changing data distribution of RL due to continual exploration that intuitively seems to hinder the identification of gradient subspaces. The limited body of studies using subspaces in PG algorithms underlines the need for a more profound discussion in this domain.

---

[*]Correspondence to `jan.schneider@tuebingen.mpg.de`

This paper conducts a comprehensive empirical evaluation of gradient subspaces in the context of PG algorithms, assessing their properties across various simulated RL benchmarks. Our experiments reveal several key findings: (i) there exist parameter-space directions that exhibit significantly larger curvature compared to other parameter-space directions, (ii) the gradients live in the subspace spanned by these directions, and (iii) the subspace remains relatively stable throughout the RL training. Additionally, we analyze the gradients of the critic – an integral part of the PG estimation in actor-critic methods – and observe that the critic subspace often exhibits less variability and retains a larger portion of its gradient compared to the actor subspace. We also test the robustness of PG subspaces regarding mini-batch approximations of the gradient that are used in practice during training and evaluate a similar mini-batch approximation of the Hessian. Lastly, we explore the extent to which the variation in the data distribution influences the aforementioned subspace analysis by conducting experiments with both an on-policy as well as an off-policy algorithm, the latter of which reuses previously collected data for training.

By shedding light on gradient subspaces in deep RL, this paper provides insights that can potentially enhance RL performance by advancing parameter-space exploration or enabling second-order optimization. We begin by reviewing existing literature on subspace approaches in Section 2, followed by a recapitulation of the RL preliminaries in Section 3 as a foundation for the analysis of gradient subspaces in RL in Section 4. Section 5 concludes this work with a discussion of the results and implications of our work. The code for our experiments is available on the project website.

## 2 RELATED WORK

Numerous works have studied the use of gradient subspaces in SL. These works can be roughly divided into *informed* and *random* subspace approaches. In the following, we give an overview of these papers and highlight works that investigate related concepts in RL.

**Informed subspaces in supervised learning**  Informed subspace methods identify the subspace from some part of the training process. For instance, Gur-Ari et al. (2018) show that the gradients used for neural network optimization lie in a low-dimensional subspace of the parameter-space directions with the highest curvature in the loss. Furthermore, they show that this subspace changes slowly throughout the training. Recent works have highlighted numerous benefits of these insights. Li et al. (2022a) apply principal component analysis on the network parameters at previous checkpoints and use the top principal components as subspace. They apply SGD and BFGS algorithms in the subspace and demonstrate superior learning performance compared to training in the original parameter space. Similarly, Tuddenham et al. (2020) propose to use a BFGS method in a low-dimensional subspace defined by the full-batch gradients with respect to individual classification labels. Sohl-Dickstein et al. (2014) construct a second-order Taylor approximation of individual loss terms efficiently in a low-dimensional subspace that captures recently observed gradients. With this approximation, the overall loss is a sum of quadratic functions that can be optimized in closed form. Chen et al. (2022) utilize insights about the low-dimensional subspaces to learn an optimizer that scales to high-dimensional parameter spaces. Traditionally, learned optimizers are limited in scale by the high output dimensionality of the network. By outputting parameter updates in the low-dimensional subspace, Chen et al. circumvent this challenge and efficiently learn to optimize large neural networks. Gauch et al. (2022) apply the knowledge of low-dimensional gradient subspace to few-shot adaptation by identifying a suitable subspace on the training task and utilizing it for efficient finetuning on the test task. Likewise, Zhou et al. (2020) tackle differential privacy by identifying a gradient subspace on a public dataset and projecting gradients calculated from the private dataset into this subspace. Adding noise in the subspace instead of the parameter space lowers the error bound of differential privacy to be linear in the number of subspace dimensions instead of the parameter dimensions. Li et al. (2022b) tackle an overfitting problem of adversarial training by identifying a gradient subspace in an early training phase before overfitting occurs. By projecting later gradients into this subspace, they enforce that the training dynamics remain similar to the early training phase, which mitigates the overfitting problem.

**Random subspaces in supervised learning**  Gressmann et al. (2020) describe different modifications to improve neural network training in random subspaces, like resampling the subspace regularly and splitting the network into multiple components with individual subspaces. They utilize these subspaces to reduce the communication overhead in a distributed learning setting by send-

ing the parameter updates as low-dimensional subspace coefficients. Li et al. (2018) quantify the difficulty of a learning task by its *intrinsic dimensionality*, i.e., the dimensionality of the random subspace needed to reach good performance. Aghajanyan et al. (2021) apply a similar analysis to language models and find that the effectiveness of fine-tuning language models is tied to a low intrinsic dimensionality. Based on these findings, Hu et al. (2021) propose a method for fine-tuning large models efficiently via low-rank parameter updates. Larsen et al. (2021) compare informed and random subspaces and find that with informed subspaces, typically significantly fewer dimensions are needed for training. Furthermore, they note the benefit of first training in the original space for some time before starting subspace training, as this increases the probability of the subspace intersecting with low-loss regions in the parameter space.

**Subspace approaches to reinforcement learning**  Only a handful of works use subspace methods for RL. Tang et al. (2023) identify a parameter subspace through singular value decomposition of the policy parameters' temporal evolution. They propose to cut off the components of the update step that lie outside this subspace and additionally elongate the update in the directions corresponding to the largest singular values. In contrast to our work, they focus solely on the policy network, consider only off-policy learning algorithms, and do not establish a connection to the curvature of the learning objective. Li et al. (2018) quantify the difficulty of RL tasks by their intrinsic dimensionality. To that end, the authors optimize the value function of Deep Q-Networks (DQN) (Mnih et al., 2013) as well as the population distribution – a distribution over the policy parameters – of evolutionary strategies (ES) (Salimans et al., 2017) in random subspaces. Maheswaranathan et al. (2019) define a search distribution for ES that is elongated along an informed subspace spanned by surrogate gradients. While the authors do not evaluate their approach for RL, they highlight the relevance to RL due to the method's ability to use surrogate gradients generated from learned models, similar to the use of the critic in actor-critic methods. Gaya et al. (2022) enable improved generalization in online RL by learning a subspace of policies. Recently, they extended their method to scale favorably to continual learning settings (Gaya et al., 2023). Both methods explicitly learn a subspace of policy parameters, while our work investigates the natural training dynamics of PG methods.

## 3 PRELIMINARIES

This section introduces the mathematical background and notation used throughout the paper. Furthermore, we briefly describe the two RL algorithms that we will analyze in Section 4.

### 3.1 MATHEMATICAL BACKGROUND AND NOTATION

For a given objective function $J(\theta)$, we use $g = \frac{\partial}{\partial \theta} J(\theta) \in \mathbb{R}^n$ to denote the gradient of a model with respect to its parameters $\theta$ and $H = \frac{\partial^2}{\partial^2 \theta} J(\theta) \in \mathbb{R}^{n \times n}$ to denote the corresponding Hessian matrix. We use $v_i$ to denote the $i$th largest eigenvector of $H$. Note that we use "$i$th largest eigenvector" as shorthand for "eigenvector with respect to the $i$th largest eigenvalue". Since $H$ is symmetric, all eigenvectors are orthogonal to each other, and we assume $||v_i|| = 1$.

In this work, we investigate projections of gradients into lower-dimensional subspaces, i.e., mappings from $\mathbb{R}^n$ to $\mathbb{R}^k$ with $k \ll n$. These mappings are defined by a *projection matrix* $P \in \mathbb{R}^{k \times n}$. $\hat{g} = P g \in \mathbb{R}^k$ denotes the projection of $g$ into the subspace and $\tilde{g} = P^+ \hat{g} \in \mathbb{R}^n$ is the mapping of $\hat{g}$ back to the original dimensionality that minimizes the projection error $||g - \tilde{g}||$. Here, $P^+$ denotes the pseudoinverse of $P$. If the projection matrix is semi-orthogonal, i.e., the columns are orthogonal and norm one, the pseudoinverse simplifies to the transpose $P^+ = P^T$. The matrix of the $k$ largest eigenvectors $P = (v_1, \dots, v_k)^T$ is one example of such a semi-orthogonal matrix.

### 3.2 REINFORCEMENT LEARNING

We consider tasks formulated as Markov decision processes (MDPs), defined by the tuple $(\mathcal{S}, \mathcal{A}, p, r)$. Here, $\mathcal{S}$ and $\mathcal{A}$ are the state and action spaces, respectively. The transition dynamics $p \colon \mathcal{S} \times \mathcal{A} \times \mathcal{S} \to [0, \infty)$ define the probability density of evolving from one state to another. At each timestep the agent receives a scalar reward $r \colon \mathcal{S} \times \mathcal{A} \to \mathbb{R}$. A stochastic policy, $\pi_\theta(\mathbf{a}_t \mid \mathbf{s}_t)$, defines a mapping from state $\mathbf{s}_t$ to a probability distribution over actions $\mathbf{a}_t$. RL aims to find an optimal policy $\pi^*$, maximizing the expected cumulative return, discounted by $\gamma \in [0, 1]$.

The value function $V^\pi(\mathbf{s})$ represents the expected (discounted) cumulative reward from state $\mathbf{s}$ following policy $\pi$, and the action value function $Q^\pi(\mathbf{s}, \mathbf{a})$ denotes the expected (discounted) cumulative reward for taking action $\mathbf{a}$ in state $\mathbf{s}$ and then following $\pi$. The advantage function $A^\pi(\mathbf{s}, \mathbf{a}) = Q^\pi(\mathbf{s}, \mathbf{a}) - V^\pi(\mathbf{s})$ quantifies the relative benefit of taking an action $\mathbf{a}$ in state $\mathbf{s}$ over the average action according to policy $\pi$.

RL algorithms generally can be divided into two styles of learning. On-policy methods, like Proximal Policy Optimization (PPO) (Schulman et al., 2017), only use data generated from the current policy for updates. In contrast, off-policy algorithms, such as Soft Actor-Critic (SAC) (Haarnoja et al., 2018) leverage data collected from different policies, such as old iterations of the policy.

### 3.2.1 PROXIMAL POLICY OPTIMIZATION

On-policy PG methods typically optimize the policy $\pi_\theta$ via an objective such as $J(\theta) = \mathbb{E}_{\pi_\theta}\left[\pi_\theta(\mathbf{a}_t \mid \mathbf{s}_t)\hat{A}_t\right] = \mathbb{E}_{\pi_{\mathrm{old}}}\left[\mathrm{r}_t(\theta)\hat{A}_t\right]$ with $\mathrm{r}_t(\theta) = \frac{\pi_\theta(\mathbf{a}_t|\mathbf{s}_t)}{\pi_{\mathrm{old}}(\mathbf{a}_t|\mathbf{s}_t)}$ and $\hat{A}_t$ being an estimator of the advantage function at timestep $t$ and $\pi_{\mathrm{old}}$ denoting the policy before the update (Kakade & Langford, 2002). However, optimizing this objective can result in excessively large updates, leading to instabilities and possibly divergence. Proximal Policy Optimization (PPO) (Schulman et al., 2017) is an on-policy actor-critic algorithm designed to address this issue by clipping the probability ratio $r_t(\theta)$ to the interval $[1-\epsilon, 1+\epsilon]$, which removes the incentive for moving $\mathrm{r}_t(\theta)$ outside the interval, resulting in the following actor loss.

$$J_{\mathrm{actor}}^{\mathrm{PPO}}(\theta) = \mathbb{E}_{\pi_{\mathrm{old}}}\left[\min\left(\mathrm{r}_t(\theta)\hat{A}_t, \mathrm{clip}\left(\mathrm{r}_t(\theta), 1-\epsilon, 1+\epsilon\right)\hat{A}_t\right)\right] \tag{1}$$

The advantage estimation $\hat{A}_t = \delta_t + (\gamma\lambda)\delta_{t+1} + \cdots + (\gamma\lambda)^{T-t+1}\delta_{T-1}$ with $\delta_t = r_t + \gamma V_\phi(\mathbf{s}_{t+1}) - V_\phi(\mathbf{s}_t)$ uses a learned value function $V_\phi$, which acts as a *critic*. The hyperparameter $\lambda$ determines the trade-off between observed rewards and estimated values. The critic is trained to minimize the mean squared error between the predicted value and the discounted sum of future episode rewards $V_t^{\mathrm{target}} = \sum_{\tau=t}^{t+T}\gamma^{\tau-t}r_\tau$.

$$J_{\mathrm{critic}}^{\mathrm{PPO}}(\phi) = \mathbb{E}\left[(V_\phi(\mathbf{s}_t) - V_t^{\mathrm{target}})^2\right] \tag{2}$$

### 3.2.2 SOFT ACTOR-CRITIC

Soft Actor-Critic (SAC) (Haarnoja et al., 2018) is a policy gradient algorithm that integrates the maximum entropy reinforcement learning framework with the actor-critic approach. As such, it optimizes a trade-off between the expected return and the policy's entropy. It is an off-policy algorithm and, as such, stores transitions in a replay buffer $\mathcal{D}$, which it samples from during optimization.

To that end, SAC modifies the targets for the learned Q-function $Q_\phi$ to include a term that incentivizes policies with large entropy $\hat{Q}_t = r_t + \gamma(Q_\phi(\mathbf{s}_{t+1}, \mathbf{a}_{t+1}) - \log\pi_\theta(\mathbf{a}_{t+1} \mid \mathbf{s}_{t+1}))$, resulting in the following critic loss.

$$J_{\mathrm{critic}}^{\mathrm{SAC}}(\phi) = \mathbb{E}_{\mathcal{D},\pi_\theta}\left[\frac{1}{2}\left(Q_\phi(\mathbf{s}_t, \mathbf{a}_t) - \hat{Q}_t\right)^2\right] \tag{3}$$

Note that SAC, in its original formulation, trains an additional value function and a second Q-function, but we omitted these details for brevity. The algorithm then trains the actor to minimize the KL-divergence between the policy and the exponential of the learned Q-function.

$$J_{\mathrm{actor}}^{\mathrm{SAC}}(\theta) = \mathbb{E}_{\mathcal{D}}\left[D_{\mathrm{KL}}\left(\pi_\theta(\cdot \mid \mathbf{s}_t) \left\| \frac{\exp(Q_\phi(\mathbf{s}_t, \cdot))}{Z_\phi(\mathbf{s}_t)}\right.\right)\right] \tag{4}$$

$Z_\phi(\mathbf{s}_t)$ denotes the normalization to make the right side of the KL-divergence a proper distribution. Optimizing this loss increases the probability of actions with high value under the Q-function.

## 4 GRADIENT SUBSPACES IN POLICY GRADIENT ALGORITHMS

In Section 2, we have highlighted several works from SL that utilize low-dimensional gradient subspaces for improving the learning performance. Naturally, we would like to transfer these benefits

to policy gradient algorithms. However, the training in RL is significantly less stationary than in the supervised setting (Bjorck et al., 2022). As the RL agent changes, the data distribution shifts since the data is generated by the agent's interactions with its environment. Furthermore, the value of a state also depends on the agent's behavior in future states. Thus, the targets for the actor and critic networks change constantly. These crucial differences between SL and RL underscore the need to analyze to which extent insights about gradient subspaces transfer between these settings.

The analysis presented in this section focuses on two policy gradient algorithms: PPO (Schulman et al., 2017) and SAC (Haarnoja et al., 2018), which are popular instantiations of on-policy and off-policy RL. We apply the algorithms to twelve benchmark tasks from OpenAI Gym (Brockman et al., 2016), Gym Robotics (Plappert et al., 2018a), and the DeepMind Control Suite (Tunyasuvunakool et al., 2020). Our code builds upon the algorithm implementations of *Stable Baselines3* (Raffin et al., 2021). The learning curves are displayed in Appendix A. We ran each experiment for 10 random seeds and plot the mean and standard deviation for the results in Sections 4.2 and 4.3. Due to space constraints, we show the analysis results only for selected tasks and present detailed results for all twelve tasks in Appendix B. Moreover, we conduct an evaluation of the impact of the RL algorithm's hyperparameters on the gradient subspace in Appendix C.

For the following analyses, we calculate Hessian eigenvectors of the loss with respect to the network parameters via the Lanczos method (Lehoucq et al., 1998) since it is an efficient method for estimating the top eigenvectors that avoids explicitly constructing the Hessian matrix. Since we can only estimate the Hessian from data, we use a large number of state-action pairs to obtain precise estimates for the eigenvectors of the true Hessian, similar to how Ilyas et al. (2020) approximate the true policy gradient. For PPO, we collect $10^6$ on-policy samples. This would, however, not be faithful to the diverse distribution of off-policy data that SAC uses for training. To match this data distribution for the analysis, we save the replay buffer during training and use the data of the complete replay buffer for estimating the Hessian. Note that the replay buffer also has a capacity of $10^6$ samples but is not completely filled at the beginning of training.

As mentioned in Section 3, SAC and PPO each train two different networks, an actor and a critic. We, therefore, conduct our analysis for each network individually. To verify that there exist high-curvature directions spanning a subspace that stays relatively stable throughout the training and that contains the gradient, we check three conditions:

   i) Some parameter-space directions exhibit significantly larger curvature in the actor/critic loss than other directions.
  ii) The actor/critic gradient mainly lies in the subspace spanned by these directions.
 iii) The subspaces for the actor and critic networks change slowly throughout the training.

### 4.1 THE LOSS CURVATURE IS LARGE IN A FEW PARAMETER-SPACE DIRECTIONS

The Hessian matrix describes the curvature of a function, with the eigenvectors being the directions of maximum and minimum curvature. The corresponding eigenvalues describe the magnitude of the curvature along these directions. Therefore, we verify condition i) by plotting the spectrum of Hessian eigenvalues for the actor and critic loss of PPO with respect to the network parameters in Figure 1. The plots show that there are a few large eigenvalues for both the actor and critic loss. All remaining eigenvalues are distributed close to zero. These plots confirm that there are a few directions with significantly larger curvature; in other words, the problem is ill-conditioned.

### 4.2 THE GRADIENT LIES IN THE HIGH-CURVATURE SUBSPACE

To verify condition ii) that the high-curvature subspace contains the gradients of the respective loss, we measure how well these gradients can be represented in the subspace. Let $P_k \in \mathbb{R}^{k \times n}$ be the semi-orthogonal matrix that projects into the high-curvature subspace. $P_k$ consists row-wise of the $k$ largest Hessian eigenvectors. We compute the relative projection error, i.e., the relative difference between the original gradient $g \in \mathbb{R}^n$ and the *projected gradient* $\tilde{g} = P_k^+ P_k g$ that is the result of mapping into the high-curvature subspace and back into the original space. The fraction of the gradient that can be represented in the subspace is then given by

$$S_{\text{frac}}(P_k, g) = 1 - \frac{||\tilde{g} - g||^2}{||g||^2}, \tag{5}$$

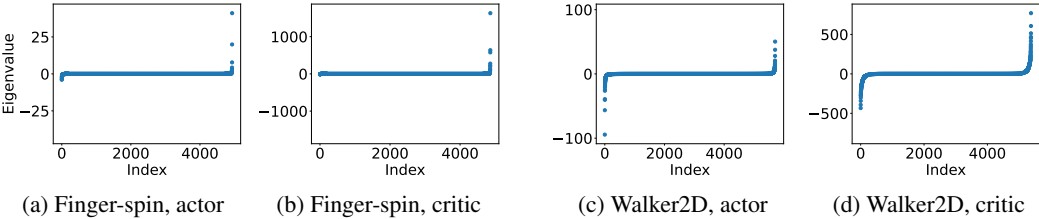

| (a) Finger-spin, actor | (b) Finger-spin, critic | (c) Walker2D, actor | (d) Walker2D, critic |

Figure 1: The spectrum of the Hessian eigenvalues for PPO on the tasks Finger-spin (a, b) and Walker2D (c, d). The Hessian is estimated from $10^6$ state-action pairs. For both the actor (a, c) and critic (b, d) loss, there is a small number of large eigenvalues, while the bulk of the eigenvalues is close to zero. This finding shows that there is a small number of high-curvature directions in the loss landscapes, which is in accordance with results from SL.

which simplifies to the following *gradient subspace fraction* criterion of Gur-Ari et al. (2018).

$$S_{\text{frac}}(P_k, g) = \frac{||P_k\, g||^2}{||g||^2} \tag{6}$$

We derive this equality in Appendix D. Note that $0 \leq S_{\text{frac}}(P_k, g) \leq 1$ holds, where $S_{\text{frac}}(P_k, g) = 1$ implies that the subspace perfectly contains the gradient, while $S_{\text{frac}}(P_k, g) = 0$ means that the gradient lies entirely outside the subspace. Due to the normalization by $||g||$, this criterion is invariant to the scale of the gradient, which enables comparing gradient subspaces of different models and models at different stages of the training.

To evaluate how the gradient subspace fraction evolves over time, we evaluate the criterion at checkpoints every $50,000$ steps during the RL training. To compactly visualize this data, we split the training into three phases: *initial*, *training*, and *convergence*, and for each phase, we average the results of all timesteps of that phase. Since the algorithms require different numbers of timesteps for solving each of the tasks and reach different performance levels, we define the following heuristic criterion for the training phases. We first smooth the learning curves by averaging over a sliding window and compute the maximum episode return $R_{\max}$ over the smoothed curve. Next, we calculate the improvement relative to the episode return $R_{\text{init}}$ of the initial policy at each timestep $t$ of the smoothed learning curve as

$$\Delta_R(t) = \frac{R(t) - R_{\text{init}}}{R_{\max} - R_{\text{init}}}. \tag{7}$$

We then define the end of the initial phase as the first timestep at which the agent reaches 10% of the total improvement, i.e., $\Delta_R(t) \geq 0.1$. Similarly, we define the start of the convergence phase as the first timestep at which the agent reaches 90% of the total improvement, i.e., $\Delta_R(t) \geq 0.9$.

We choose $k = 100$ as subspace dimensionality since this subspace already largely captures the gradients, and the largest $100$ eigenvectors can still be calculated reasonably efficiently with the Lanczos method. Appendix B displays results for different subspace sizes. With the tuned hyperparameters from *RL Baselines3 Zoo* that we use for training, the PPO actor and critic usually contain around $5,000$ parameters, and the SAC actor and critic around $70,000$ and $140,000$ parameters (2 Q-networks à $70,000$ parameters), respectively. Hence, the subspace dimensionality is around 2% the size of the parameters for PPO and around 0.14% and 0.07% for SAC.

We consider a precise approximation of the true gradient computed with $10^6$ state-action pairs for PPO and the full replay buffer for SAC. We denote this approximation as *precise gradient* and the low-sample gradient used during regular RL training as *mini-batch gradient*. In a similar manner, we denote the Hessian estimated on the large dataset as *precise Hessian* and the estimate from $2048$ samples as *mini-batch Hessian*. We choose $2048$ samples for the mini-batch Hessian since that is the amount of data that PPO with default hyperparameters collects for the policy updates. Hence, this is a realistic setting for estimating the subspace during training.

Figure 2 shows the value of the gradient subspace fraction $S_{\text{frac}}(P_k, g)$ for PPO and SAC on four different tasks, divided into the three training phases. Note that for an uninformed random projection, the gradient subspace fraction would be $k/n$ in expectation, i.e., the ratio of the original

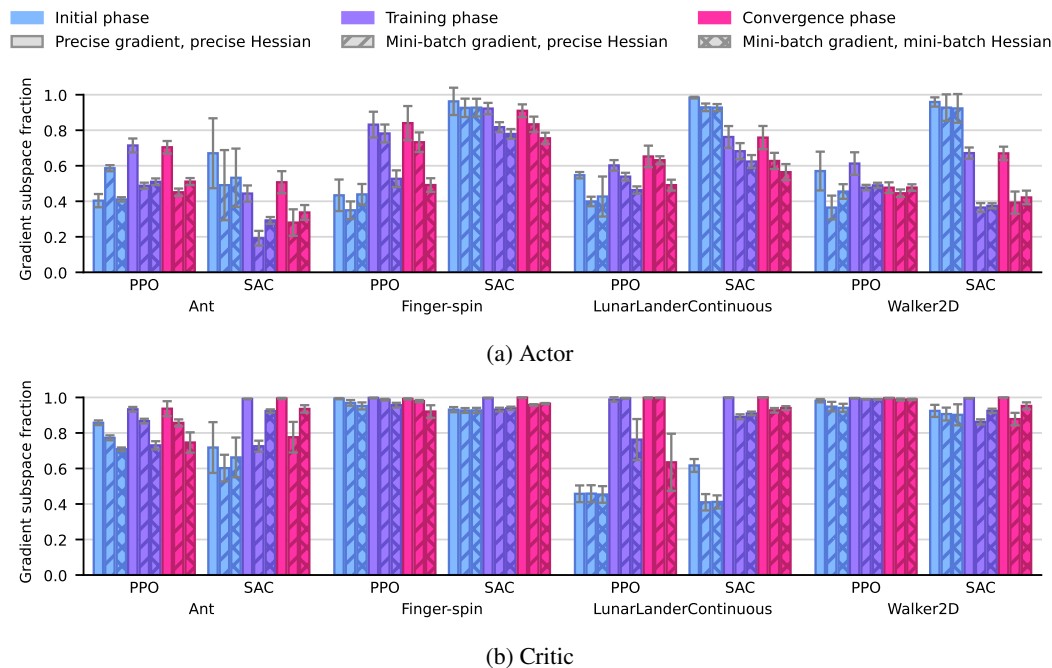

(a) Actor

(b) Critic

Figure 2: The fraction $S_{\mathrm{frac}}$ of the gradient that lies within the high-curvature subspace spanned by the 100 largest Hessian eigenvectors. Results are displayed for the actor (top) and critic (bottom) of PPO and SAC on the Ant, Finger-spin, LunarLanderContinuous, and Walker2D tasks. The results demonstrate that a significant fraction of the gradient lies within the high-curvature subspace, but the extent to which the gradient is contained in the subspace depends on the algorithm, task, and training phase. For both algorithms, the gradient subspace fraction is significantly higher for the critic than for the actor. Furthermore, the quantity is also often larger for SAC's actor than for PPO's, particularly in the early stages of the training. Even with mini-batch estimates for the gradient and Hessian, the gradient subspace fraction is considerable.

and subspace dimensionalities (0.02 for PPO and 0.0014 for SAC's actor and 0.0007 for its critic). The results in Figure 2 show a significantly higher gradient subspace fraction, which means that the gradients computed by PPO and SAC lie to a large extent in the high-curvature subspace. We observe that the fraction of the gradient in the subspace is considerably higher for the critic than for the actor. Furthermore, the gradient subspace fraction is also often higher for SAC's actor than for PPO's. This finding is particularly significant since the subspace size corresponds to a significantly lower percentage of the parameter dimensionality for SAC than for PPO. We hypothesize that the effect is caused by the off-policy nature of SAC. In the off-policy setting, the training distribution for the networks changes slowly since the optimization reuses previous data. In this regard, SAC is closer than PPO to the supervised learning setting, where the data distribution is fixed and for which Gur-Ari et al. (2018) report high gradient subspace fractions. Still, the subspace fraction for PPO is significant, considering that the dimensionality of the subspace is merely 2% of the original parameter space. Furthermore, for PPO, the subspace fraction often improves after the initial phase. Similarly, Gur-Ari et al. (2018) report for the supervised learning setting that the gradient starts evolving in the subspace only after some initial steps. However, for the SAC actor, this trend appears to be reversed, with the gradient subspace fraction being highest in the initial steps.

Moreover, the precise gradient, computed with a large number of samples, tends to lie better in the subspace than the mini-batch gradient. The noise resulting from the low-sample approximation seems to perturb the gradient out of the subspace. However, since the difference is typically small, the gradient subspace is still valid for the low-sample gradient estimates used during RL training. Lastly, even the subspace identified with the mini-batch Hessian captures the gradient to a significant extent. This property is crucial since it implies that we do not need access to the precise Hessian, which is costly to compute and might require additional data. Instead, we can already obtain a reasonable gradient subspace from the mini-batch Hessian.

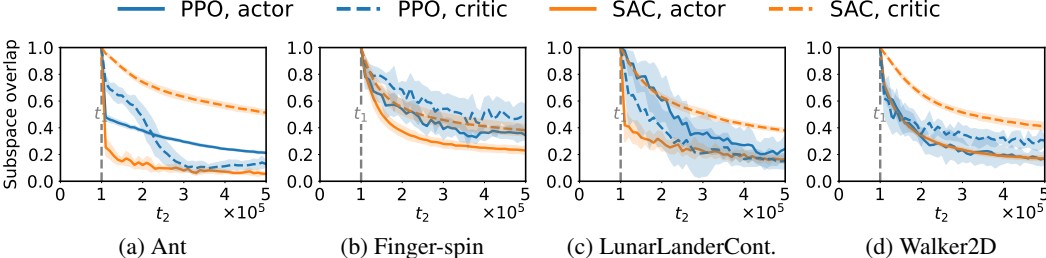

Figure 3: Evolution of the overlap between the high-curvature subspace identified at an early timestep $t_1 = 100,000$ and later timesteps for the actor and critic of PPO and SAC. While the overlap between the subspaces degrades as the networks are updated, it remains considerable even after 400,000 timesteps, indicating that the subspace remains similar, even under significant changes in the network parameters and the data distribution. This finding implies that information about the gradient subspace at earlier timesteps can be reused at later timesteps.

### 4.3 THE HIGH-CURVATURE SUBSPACE CHANGES SLOWLY THROUGHOUT THE TRAINING

So far, we have verified that the gradients of the actor and critic losses optimized by PPO and SAC lie to a large extent in the subspace spanned by the top eigenvectors of the Hessian with respect to the current parameters. However, even though there are relatively efficient methods for computing the top Hessian eigenvectors without explicitly constructing the Hessian matrix, calculating these eigenvectors at every step would be computationally expensive. Ideally, we would like to identify a subspace once that remains constant throughout the training. In practice, however, the gradient subspace will not stay exactly the same during the training, but if it changes slowly, it is possible to reuse knowledge from earlier timesteps and update the subspace at a lower frequency. To that end, we investigate condition iii) by calculating the *subspace overlap*, defined by Gur-Ari et al. (2018). The subspace overlap between timesteps $t_1$ and $t_2$ is defined as

$$S_{\text{overlap}}\left( P_k^{(t_1)}, P_k^{(t_2)} \right) = \frac{1}{k} \sum_{i=1}^{k} \left\| P_k^{(t_1)} v_i^{(t_2)} \right\|^2, \tag{8}$$

where $v_i^{(t)}$ is the $i$th largest eigenvector at timestep $t$. $P_k^{(t)} = \left( v_1^{(t)}, \ldots, v_k^{(t)} \right)^T$ denotes the projection matrix from the full parameter space to the high-curvature subspace, identified at timestep $t$. Similar to Equation (5), the criterion measures how much of the original vector is preserved during the projection into the subspace. For the subspace overlap, however, we use the projection matrix at timestep $t_1$ not to project the gradient but rather project the Hessian eigenvectors that span the high-curvature subspace identified at a later timestep $t_2$ of the training. This criterion, thus, measures how much the gradient subspace changes between these timesteps. Note that we assume the eigenvectors $v_i^{(t)}$ to be normalized to one and therefore do not normalize by their length.

Gur-Ari et al. (2018) showed in the supervised setting that the gradient subspace stabilizes only after some initial update steps. Therefore, we choose the timestep $t_1$ at which we initially identify the subspace as $t_1 = 100,000$ since this is still relatively early in the training, but the gradient subspace should already have stabilized reasonably well. We evaluate the subspace overlap criterion every 10,000 timesteps until timestep 500,000. This interval covers a significant portion of the training and showcases the extent to which the subspace changes under significant differences in the network parameters and the data distribution. For the sake of completeness and to further highlight the influence of the data distribution on the subspace, we showcase the subspace overlap over the entire duration of the training in Appendix E. As in Section 4.2, we use $k = 100$ as subspace dimensionality and refer to Appendix B for the evaluation of different subspace sizes. The analysis results in Figure 3 show that the subspace overlap reduces the further apart the two timesteps $t_1$ and $t_2$ are, but in all cases, the subspace overlap remains significantly above zero, implying that information of previous subspaces can be reused at later timesteps. If the two timesteps are close to each other, the overlap is considerable. Similar to the gradient subspace fraction in Section 4.2, the subspace overlap is often more pronounced for the critic than the actor, particularly for SAC.

## 5 CONCLUSION

In this work, we showed that findings from the SL literature about gradient subspaces transfer to the RL setting. Despite the continuously changing data distribution inherent to RL, the gradients of the actor and critic networks of PPO and SAC lie in a low-dimensional, slowly-changing subspace of high curvature. We demonstrated that this property holds for both on-policy and off-policy learning, even though the distribution shift in the training data is particularly severe in the on-policy setting.

### 5.1 HIGH-CURVATURE SUBSPACES EXPLAIN CLIFFS IN REWARD LANDSCAPES

Sullivan et al. (2022) investigate visualizations of the reward landscapes around policies optimized by PPO. Reward landscapes describe the resulting cumulative rewards over the space of policy parameters. They observe empirically that these landscapes feature "cliffs" in policy gradient direction. When changing the parameters in this direction, the cumulative reward increases for small steps but drops sharply beyond this increase. In random directions, these cliffs do not seem to occur.

The results from Section 4.2 constitute a likely explanation of this phenomenon. The cliffs that the authors describe can be interpreted as signs of large curvature in the reward landscape. Our analysis demonstrates that the policy gradient is prone to lie in a high-curvature direction of the policy loss. Sullivan et al. investigate the cumulative reward, which is different from the policy loss that we analyze in this work. However, one of the fundamental assumptions of policy gradient methods is that there is a strong link between the policy loss and the cumulative reward. Therefore, high curvature in the loss likely also manifests in the cumulative reward. There is no such influence for random directions, so the curvature in gradient direction is larger than in random directions.

### 5.2 POTENTIAL OF GRADIENT SUBSPACES IN REINFORCEMENT LEARNING

Leveraging properties of gradient subspaces has proven beneficial in numerous works in SL, e.g., (Li et al., 2022a; Chen et al., 2022; Gauch et al., 2022; Zhou et al., 2020; Li et al., 2022b). The analyses in this paper demonstrate that similar subspaces can be found in popular policy gradient algorithms. In the following, we outline two opportunities for harnessing the properties of gradient subspaces and bringing the discussed benefits to RL.

**Optimization in the subspace**  While the network architectures used in reinforcement learning are often small compared to the models used in other fields of machine learning, the dimensionality of the optimization problem is still considerable. Popular optimizers, like Adam (Kingma & Ba, 2014), typically rely only on gradient information, as computing the Hessian at every timestep would be computationally very demanding in high dimensions. However, in Section 4.1, we have seen that the optimization problem is ill-conditioned. Second-order methods, like Newton's method, are known to be well-suited for ill-conditioned problems (Nocedal & Wright, 1999). With the insights of this paper, it seems feasible to reduce the dimensionality of the optimization problems in RL algorithms by optimizing in the low-dimensional subspace instead of the original parameter space. The low dimensionality of the resulting optimization problems would enable computing and inverting the Hessian matrix efficiently and make second-order optimization methods feasible.

**Guiding parameter-space exploration**  The quality of the exploration actions significantly impacts the performance of RL algorithms (Amin et al., 2021). Most RL algorithms explore by applying uncorrelated noise to the actions produced by the policy. However, this often leads to inefficient exploration, particularly in over-actuated systems, where correlated actuation is crucial (Schumacher et al., 2022). A viable alternative is to apply exploration noise to the policy parameters instead (Rückstiess et al., 2010; Plappert et al., 2018b). This approach results in a more directed exploration and can be viewed as exploring strategies similar to the current policy.

In Section 4, we observed that the gradients utilized by policy gradient methods predominantly lie within a small subspace of all parameter-space directions. As typical parameter-space exploration does not consider the properties of the training gradient when inducing parameter noise, only a small fraction of it might actually push the policy parameters along directions that are relevant to the task. Considering that the optimization mostly occurs in a restricted subspace, it might be advantageous to limit exploration to these directions. Sampling parameter noise only in the high-curvature subspace constitutes one possible way of focusing exploration on informative parameter-space directions.

## 6 REPRODUCIBILITY

We applied our analyses to proven and publicly available implementations of the RL algorithms from Stable-Baselines3 (Raffin et al., 2021) on well-known, publicly available benchmark tasks from (Brockman et al., 2016; Plappert et al., 2018b; Tunyasuvunakool et al., 2020). Further experimental details like the learning curves of the algorithms and the fine-grained analysis results for the entire training are displayed in Appendices A and B, respectively. To facilitate reproducing our results, we make our code, as well as the raw analysis data, including hyperparmeter settings and model checkpoints, publically available on the project website.

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

# A    LEARNING CURVES

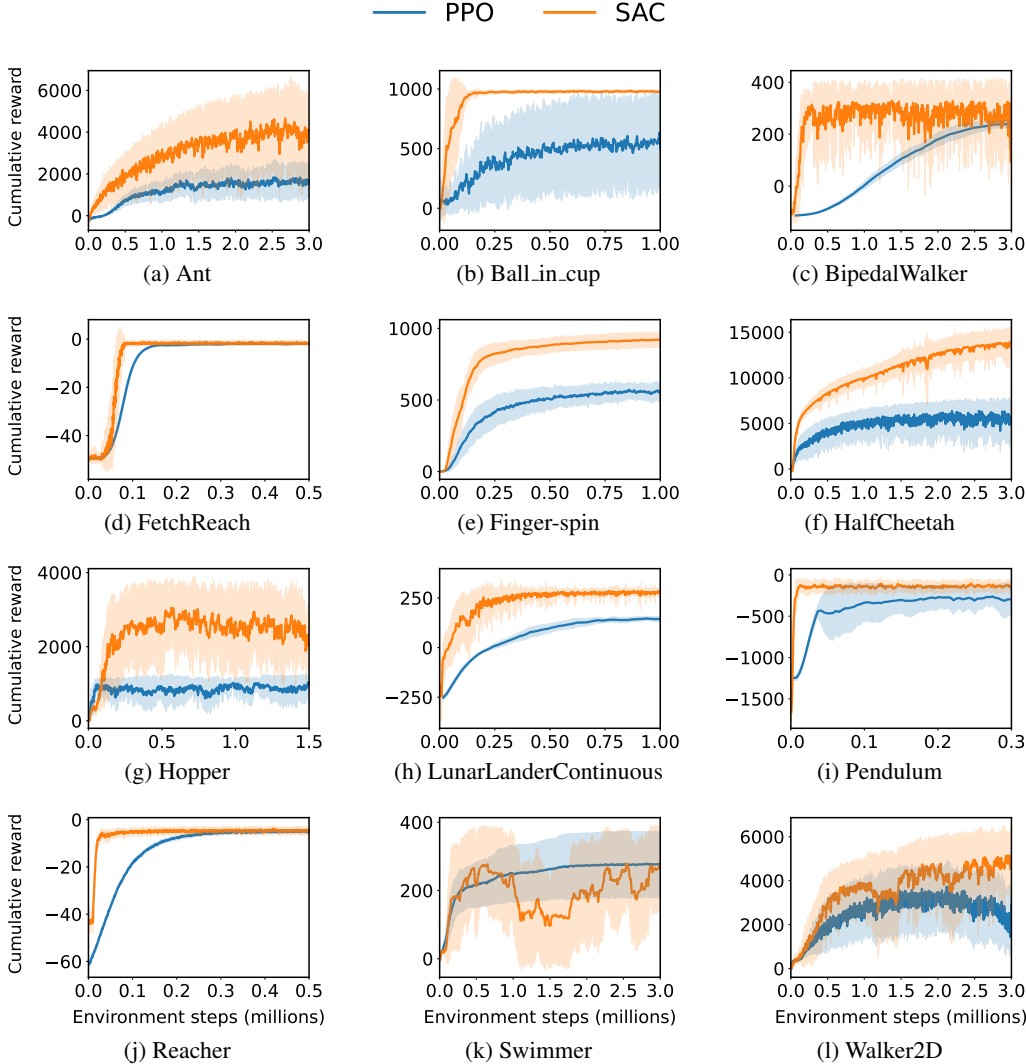

Figure 4: Learning curves for PPO and SAC on tasks from OpenAI Gym (Brockman et al., 2016), Gym Robotics (Plappert et al., 2018a), and the DeepMind Control Suite (Tunyasuvunakool et al., 2020). We use the algorithm implementations of *Stable Baselines3* (Raffin et al., 2021) with tuned hyperparameters from *RL Baselines3 Zoo* (Raffin, 2020) for the Gym tasks and hyperparameters tuned by random search over 50 configurations for the Gym Robotics and DeepMind Control Suite tasks. Results are averaged over ten random seeds; shaded areas represent the standard deviation across seeds.

# B  DETAILED ANALYSIS RESULTS FOR ALL TASKS

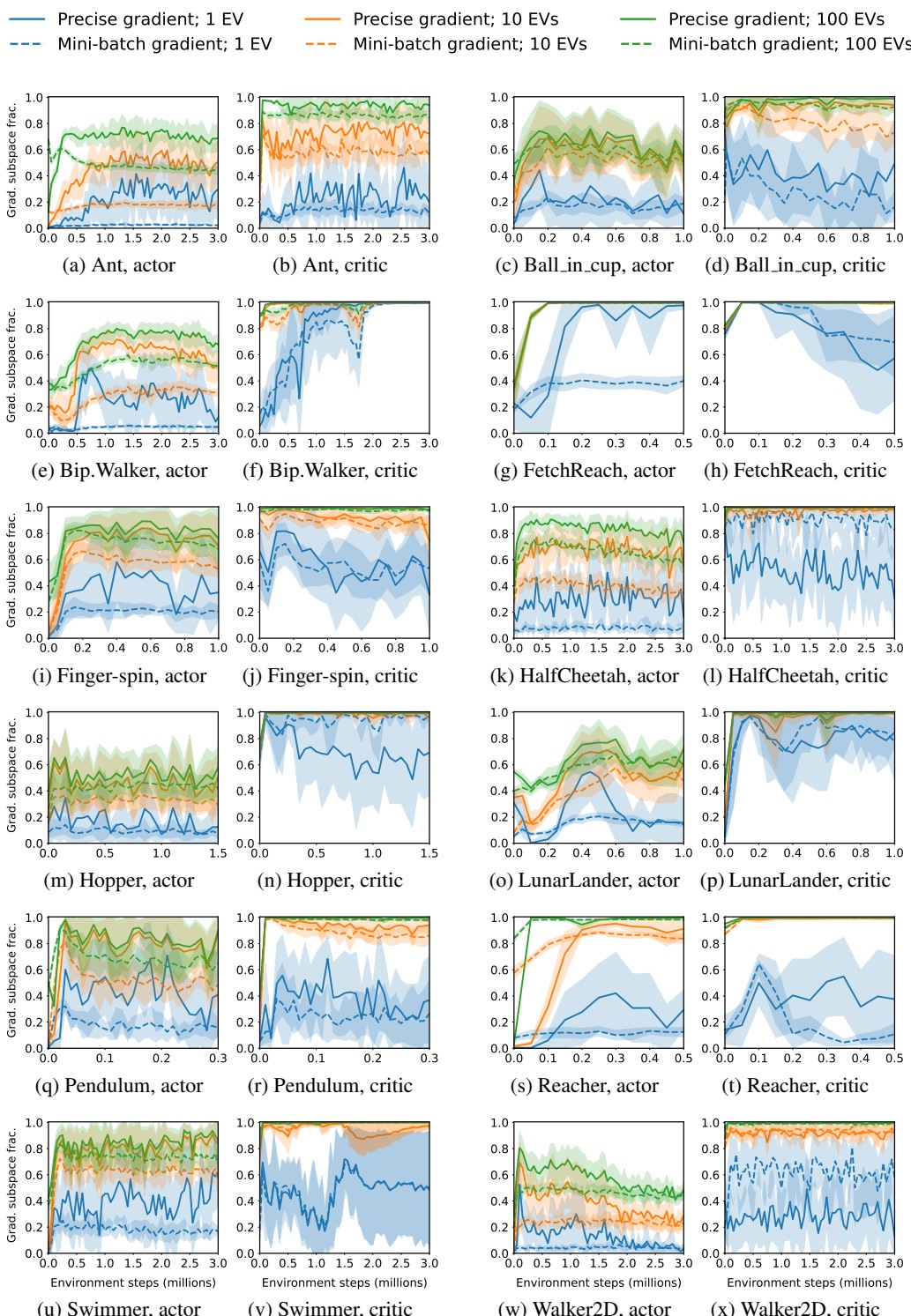

Figure 5: The evolution of the fraction of the gradient that lies within the high-curvature subspace throughout the training for PPO on all tasks. Evaluation of gradient subspaces with different numbers of eigenvectors. Results for the actor and critic.

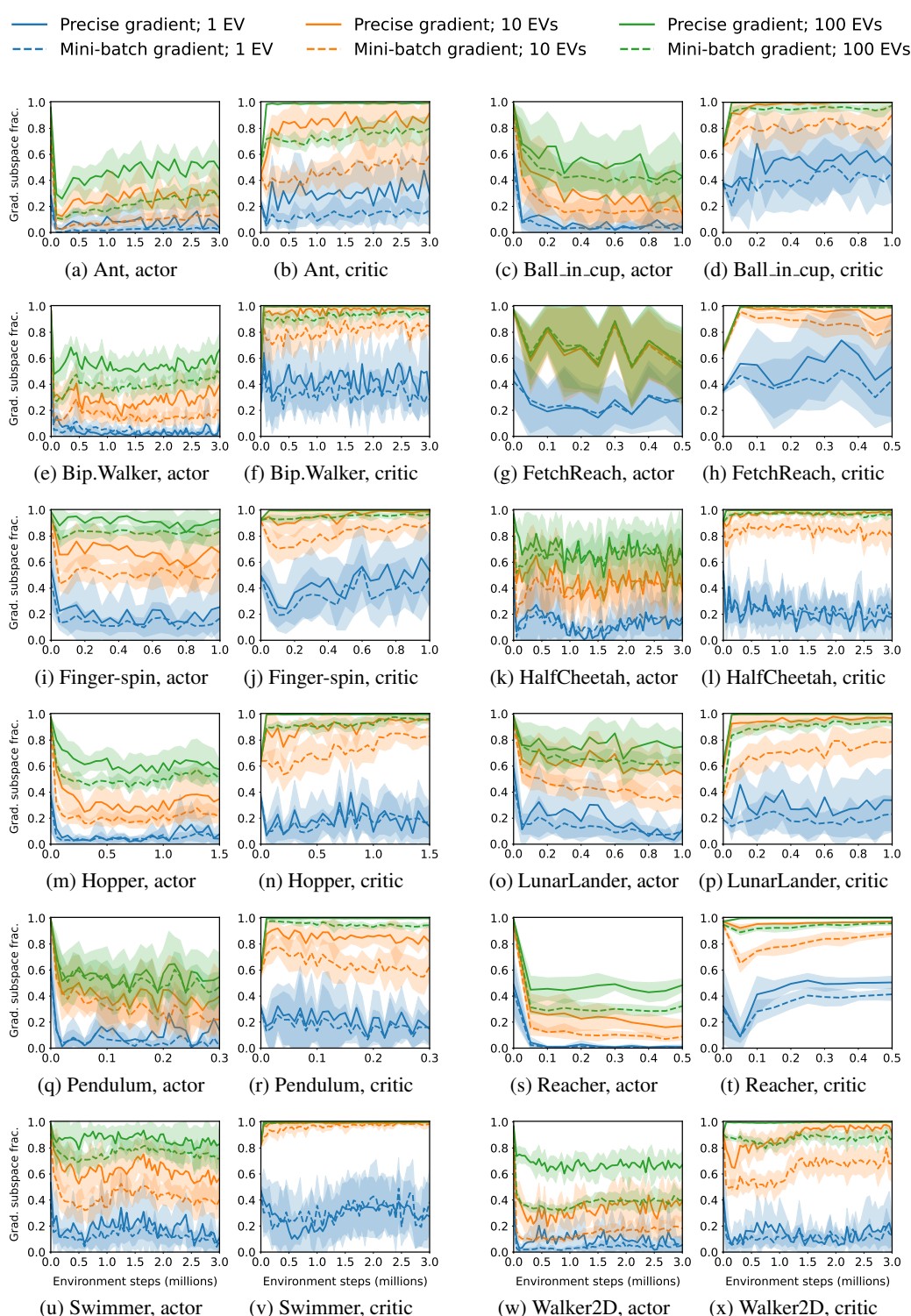

Figure 6: The evolution of the fraction of the gradient that lies within the high-curvature subspace throughout the training for SAC on all tasks. Evaluation of gradient subspaces with different numbers of eigenvectors. Results for the actor and critic.

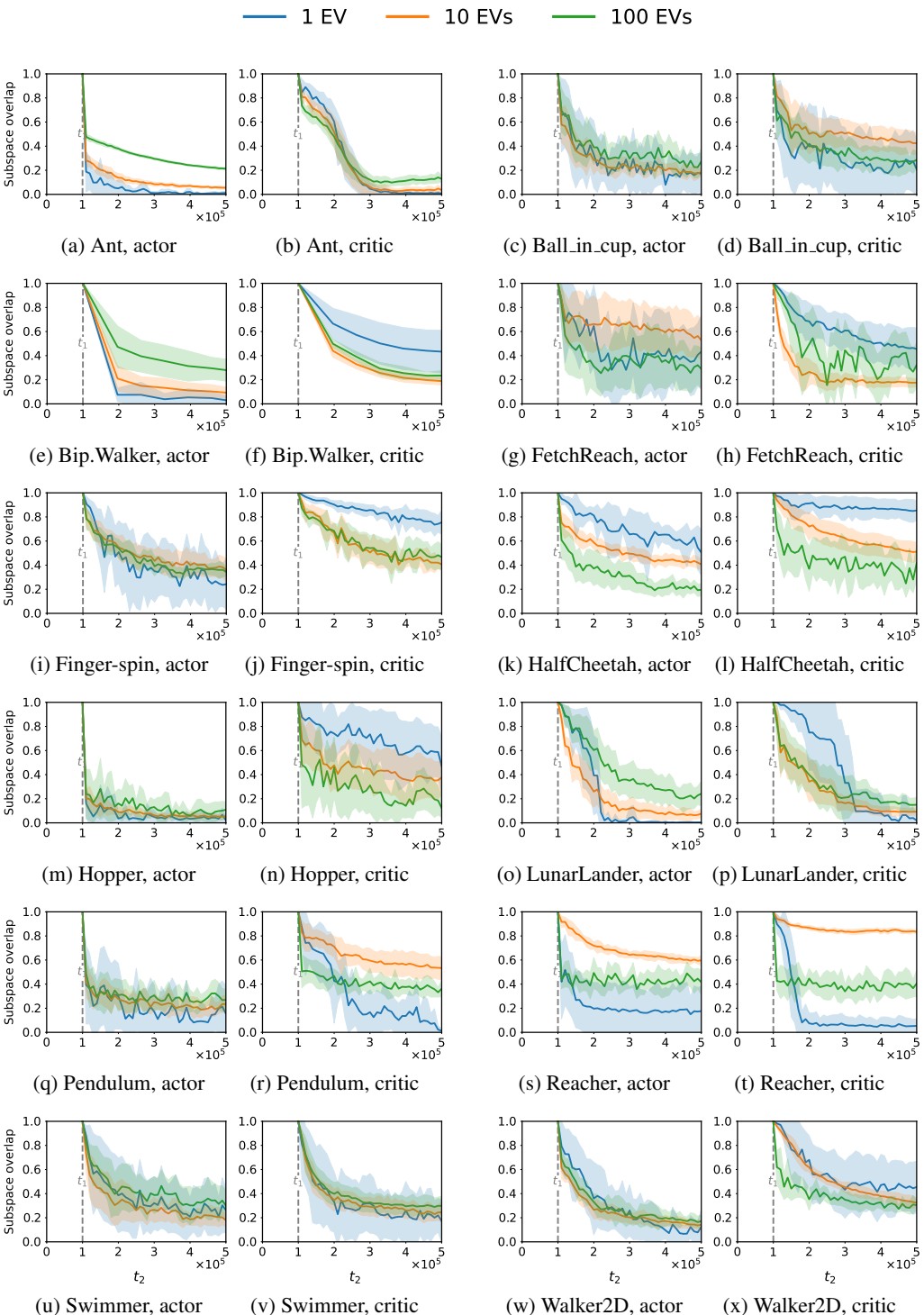

Figure 7: The evolution of the overlap between the high-curvature subspace identified at timestep $t_1 = 100,000$ and later timesteps for PPO on all tasks. Evaluation of gradient subspaces with different numbers of eigenvectors. Results for the actor and critic.

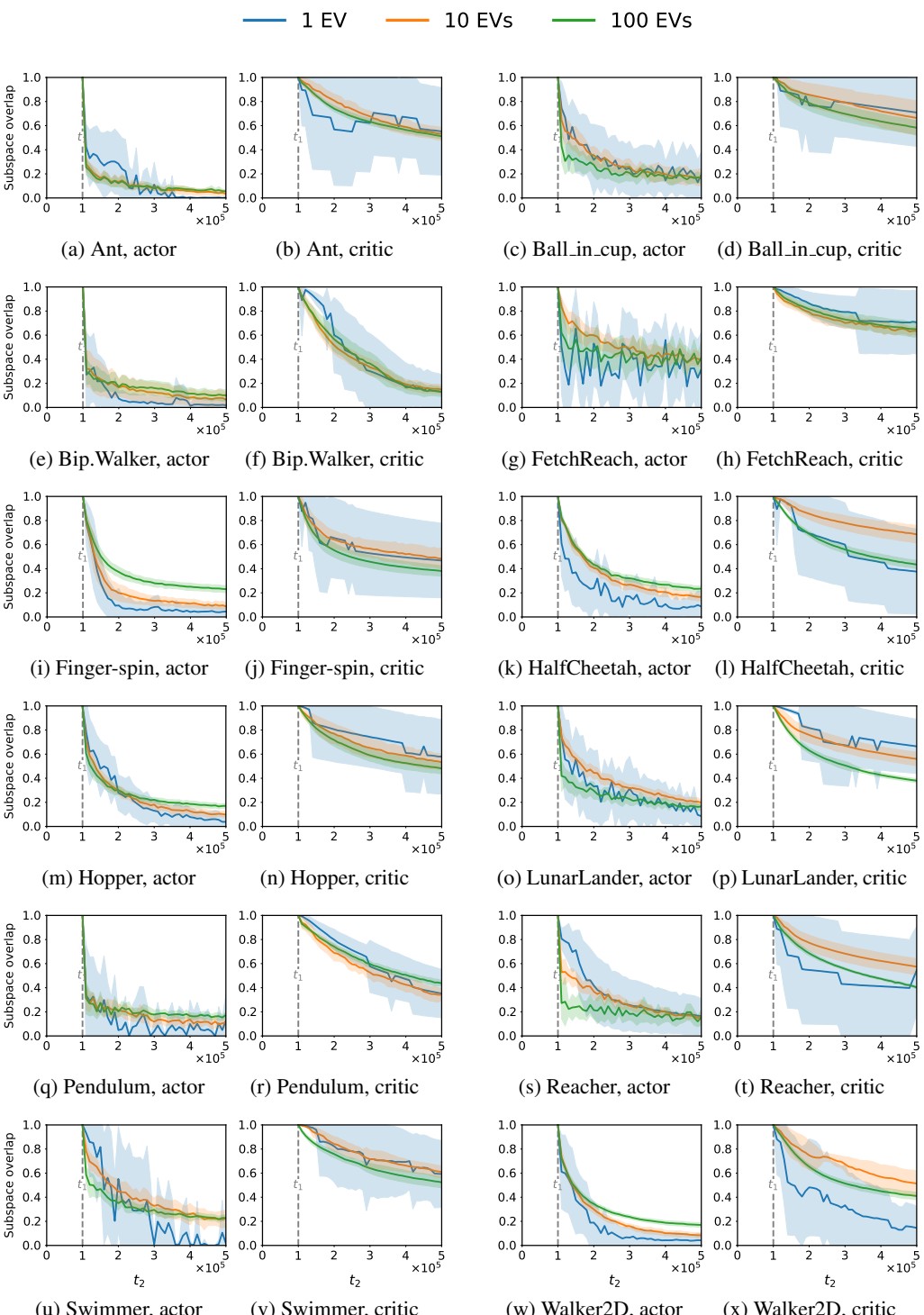

Figure 8: The evolution of the overlap between the high-curvature subspace identified at timestep $t_1 = 100{,}000$ and later timesteps for SAC on all tasks. Evaluation of gradient subspaces with different numbers of eigenvectors. Results for the actor and critic.

## C  IMPACT OF SUBOPTIMAL HYPERPARAMETERS

Hyperparameters typically have a significant impact on the learning performance of policy gradient algorithms (Paul et al., 2019). In Section 4.2, we analyzed the gradient subspace for training runs with tuned hyperparameter configurations. However, which hyperparameters work well for a given problem is typically not known a priori. Finding good hyperparameters often involves running numerous RL trainings, which might be infeasible when training on real-world tasks. For our insights to be valuable for such real-world settings, it is crucial that suitable gradient subspaces can also be identified for training runs with suboptimal hyperparameters. Therefore, in this section, we investigate the robustness of the gradient subspace with respect to suboptimal hyperparameters. To get a set of hyperparameters that are realistic but potentially suboptimal, we sample hyperparameter configurations from the ranges that are found frequently in the tuned hyperparameters of *RL Baselines3 Zoo* (Raffin, 2020). Particularly, we draw the samples from the sets given in Table 1. Note that these are also the bounds we use when sampling configurations for hyperparameter tuning.

| Algorithm | Hyperparameter | Values | Logscale |
|---|---|---|---|
| PPO | learning_rate | $[10^{-5}, 10^{-1}]$ | yes |
| | batch_size | $\{32, 64, 128, 256\}$ | no |
| | n_steps | $\{256, 512, 1024, 2048, 4096\}$ | no |
| | n_epochs | $\{5, 10, 20\}$ | no |
| | gamma | $[0.9, 1]$ | no |
| | gae_lambda | $[0.9, 1]$ | no |
| | clip_range | $\{0.1, 0.2, 0.3\}$ | no |
| | ent_coef | $[10^{-8}, 10^{-2}]$ | yes |
| | net_arch | $\{(64, 64), (128, 128), (256, 256)\}$ | no |
| SAC | learning_rate | $[10^{-5}, 10^{-1}]$ | yes |
| | batch_size | $\{32, 64, 128, 256\}$ | no |
| | train_freq | $\{1, 4, 16, 32, 64\}$ | no |
| | gamma | $[0.9, 1]$ | no |
| | tau | $\{0.005, 0.01, 0.02\}$ | no |
| | learning_starts | $\{100, 1000, 10000\}$ | no |
| | net_arch | $\{(64, 64), (128, 128), (256, 256)\}$ | no |

Table 1: Sets from which we draw random hyperparameter configurations for PPO and SAC. We refer to each hyperparameter by its name in Stable Baselines3 (Raffin et al., 2021). *Logscale* means that we first transform the interval into logspace and draw the exponent uniformly from the transformed range. All other hyperparameters are drawn uniformly. These sets reflect common hyperparameter choices from RL Baselines3 Zoo (Raffin, 2020).

We sampled a total of 100 hyperparameter configurations per algorithm and task. Figure 9 displays the distribution over the maximum cumulative reward that these hyperparameter configurations achieve. As expected, the variance in the performance is large across the hyperparameter configurations. While some configurations reach performance levels comparable to the tuned configuration, most sampled configurations converge to suboptimal behaviors.

To display the values of the gradient subspace fraction and the subspace overlap compactly for all configurations, we compress the results for each training run to a single scalar. For the gradient subspace fraction, we compute the mean of the criterion over the timesteps. Regarding the subspace overlap, we choose the early timestep $t_1 = 100{,}000$ in accordance to the experiments in Section 4.3. However, taking the mean over the remaining timesteps would not be faithful to how the subspace would typically be utilized in downstream applications. Such methods would likely update the gradient subspace after a given number of gradient steps instead of identifying it once and using it for the rest of the training. The rate of this update would likely depend on the exact way that the

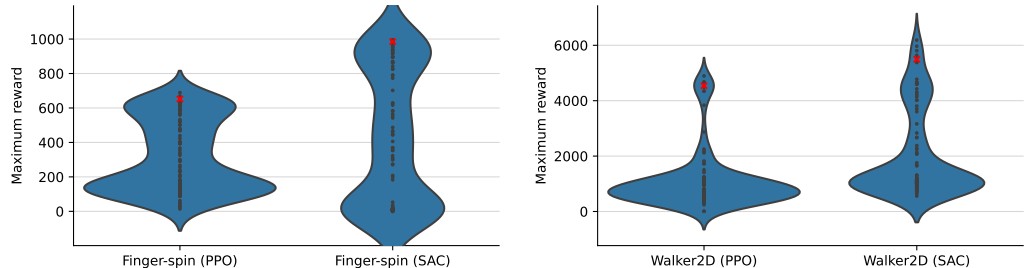

Figure 9: Violin plot of the maximum cumulative rewards achieved by 100 agents trained with random hyperparameters on Finger-spin (left) and Walker2D (right). The black dots mark the performance of the individual random configurations, and the red cross signifies the performance of the tuned hyperparameters averaged over 10 random seeds. The random hyperparameters result in agents of vastly different maximum performance. While some of the random hyperparameters achieve performance comparable to the tuned agents, the bulk of random hyperparameters stagnate at suboptimal performance levels.

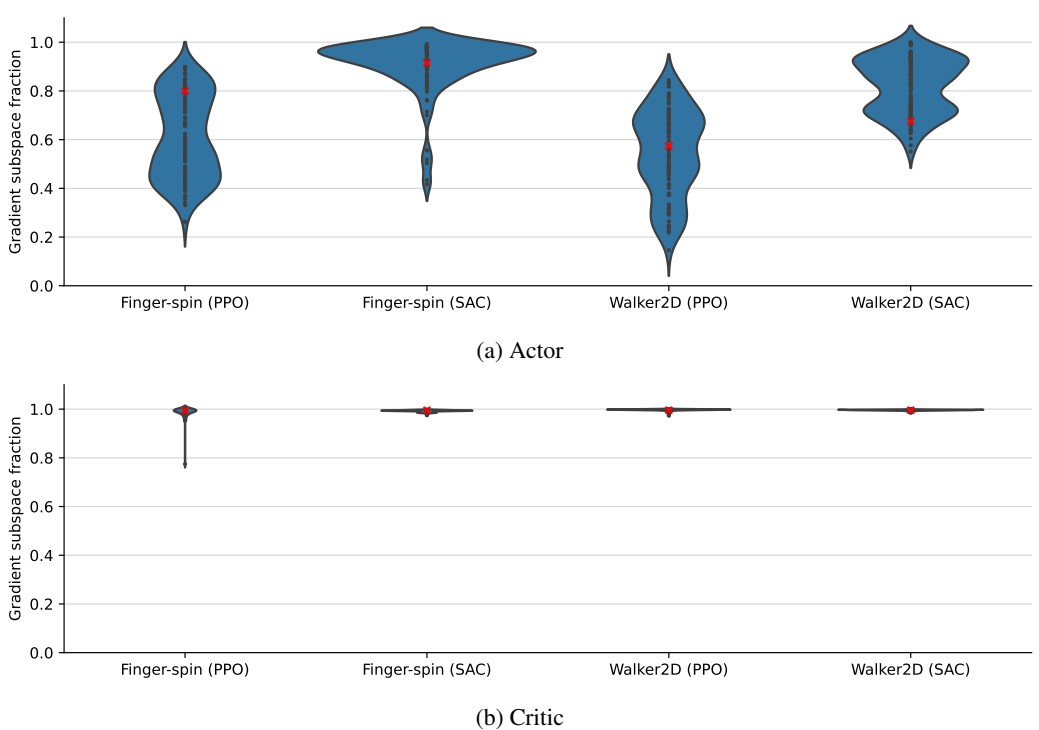

Figure 10: Violin plot of the mean gradient subspace fraction throughout the training of 100 agents with random hyperparameters. The red cross marks the same quantity for the high-performing hyperparameters used throughout the paper (averaged over 10 random seeds). For PPO's actor, the variance in the gradient subspace fraction is significantly higher than for SAC's. The critic's gradient subspace fraction is very high across all hyperparameter configurations.

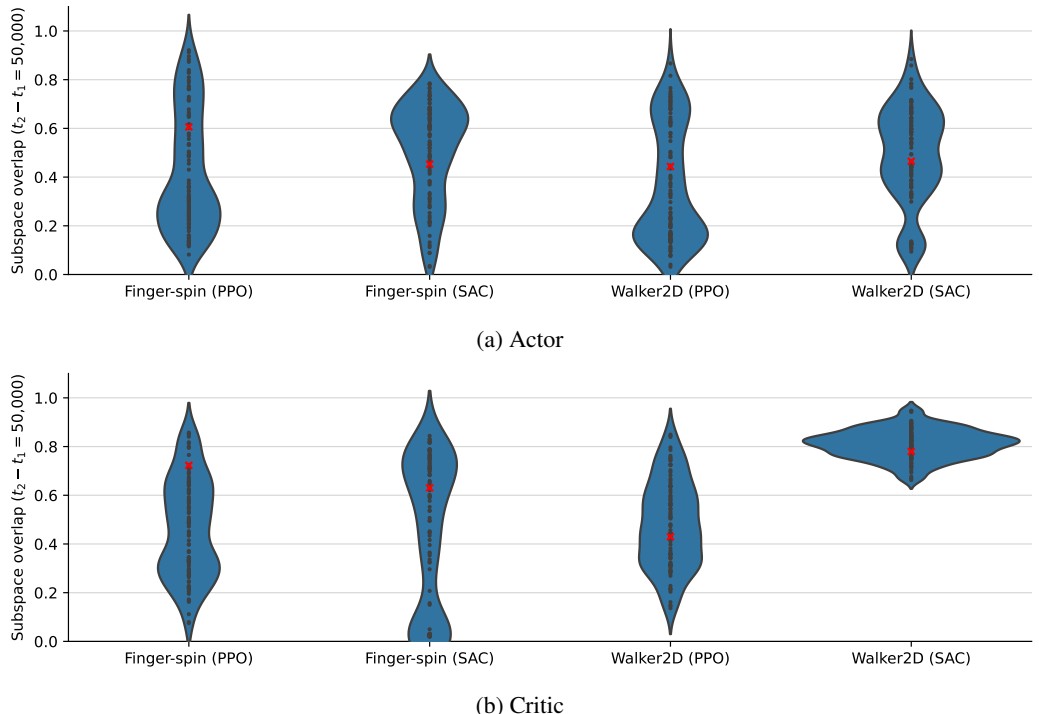

(a) Actor

(b) Critic

Figure 11: Violin plot of the mean subspace overlap for $t_1 = 100{,}000$ and $t_2 = 150{,}000$ of 50 agents with random hyperparameters. The red cross marks the same quantity for the high-performing hyperparameters used throughout the paper (averaged over 10 random seeds). For both tasks and networks, the variance of the subspace overlap is large across the hyperparameter configurations.

subspace is utilized. For this analysis, we choose $t_2 = 150{,}000$, so that the timestep difference of $t_2 - t_1 = 50{,}000$ steps is significantly shorter than the total length of the training, but the number of gradient steps is large enough that the subspace could change.

The results for the gradient subspace fraction and the subspace overlap for random hyperparameters are visualized in Figure 10 and Figure 11, respectively. Figure 10a shows for SAC's actor that the gradient is well contained in the subspace for most random hyperparameter configurations. For PPO's actor, the spread in the gradient subspace fraction is significantly higher. As shown in Figure 10b, the gradient subspace fraction for the critic is very high for all hyperparameter configurations. These results suggest that gradient subspaces can be utilized in SAC independently of the hyperparameter configuration, while PPO's actor might require more considerations.

Figure 11 shows that there is a significant spread in the subspace overlap for the random hyperparameter configurations, which indicates that potential downstream applications might need to update the gradient subspace more frequently, depending on the hyperparameters.

## D   DERIVATION OF THE GRADIENT SUBSPACE FRACTION OBJECTIVE

In this section, we derive Equation (6). Recall that $P_k = (v_1, \ldots, v_k)^T \in \mathbb{R}^{k \times n}$ is the matrix of the $k$ largest Hessian eigenvectors, which is semi-orthogonal. We use $\tilde{g} = P_k^+ P_k \, g \in \mathbb{R}^n$ to denote the projection of the gradient $g$ into the subspace and back to its original dimensionality. In the following derivation, we drop the subscript $k$ of matrix $P_k$ for ease of notation.

$$S_{\text{frac}}(P, g) = 1 - \frac{||\tilde{g} - g||^2}{||g||^2} \tag{9}$$

$$= 1 - \frac{||P^+ P g - g||^2}{||g||^2} \tag{10}$$

$$= 1 - \frac{||P^T P g - g||^2}{||g||^2} \tag{11}$$

$$= 1 - \frac{(P^T P g - g)^T (P^T P g - g)}{g^T g} \tag{12}$$

$$= 1 - \frac{(g^T P^T P - g^T)(P^T P g - g)}{g^T g} \tag{13}$$

$$= 1 - \frac{g^T P^T P P^T P g - 2 g^T P^T P g + g^T g}{g^T g} \tag{14}$$

$$= 1 - \frac{g^T P^T P g - 2 g^T P^T P g + g^T g}{g^T g} \tag{15}$$

$$= 1 - \frac{g^T g - g^T P^T P g}{g^T g} \tag{16}$$

$$= \frac{g^T g - g^T g + g^T P^T P g}{g^T g} \tag{17}$$

$$= \frac{g^T P^T P g}{g^T g} \tag{18}$$

$$= \frac{(P g)^T P g}{g^T g} \tag{19}$$

$$= \frac{||P g||^2}{||g||^2} \tag{20}$$

Note that we used the fact that the pseudo-inverse of a semi-orthogonal matrix $P$ is equal to its transpose $P^+ = P^T$ in step (11). Furthermore, we used the property $P P^T = I$ of semi-orthogonal matrices in step (15).

## E   SUBSPACE OVERLAP FOR THE ENTIRE TRAINING

In Section 4.3, we showed the subspace overlap for a range of 400,000 timesteps. Figure 12 visualizes the subspace overlap criterion with $t_1 = 100{,}000$ for all future timesteps on tasks that require training for 3 million steps. While in practical applications of gradient subspaces, the subspace would likely be updated multiple times during training, this visualization highlights the influence of the data distribution on the subspace. The plots show a small drop in the subspace overlap for SAC at 1.1 million steps. Since the replay buffer has a size of 1 million samples, this marks the point at which the original data from timestep $t_1$ is completely replaced by new data collected by updated policies. Since the networks' training data is sampled from the replay buffer, this drop indicates that this change in the data distribution has a negative effect on the subspace overlap. The effect is generally more pronounced for the critic than the actor because the actor's subspace overlap degrades faster and is already at a relatively low level at the mark of 1.1 million timesteps. For PPO, there is no such drop in the subspace overlap since the algorithm does not use experience replay and instead collects new data for every update.

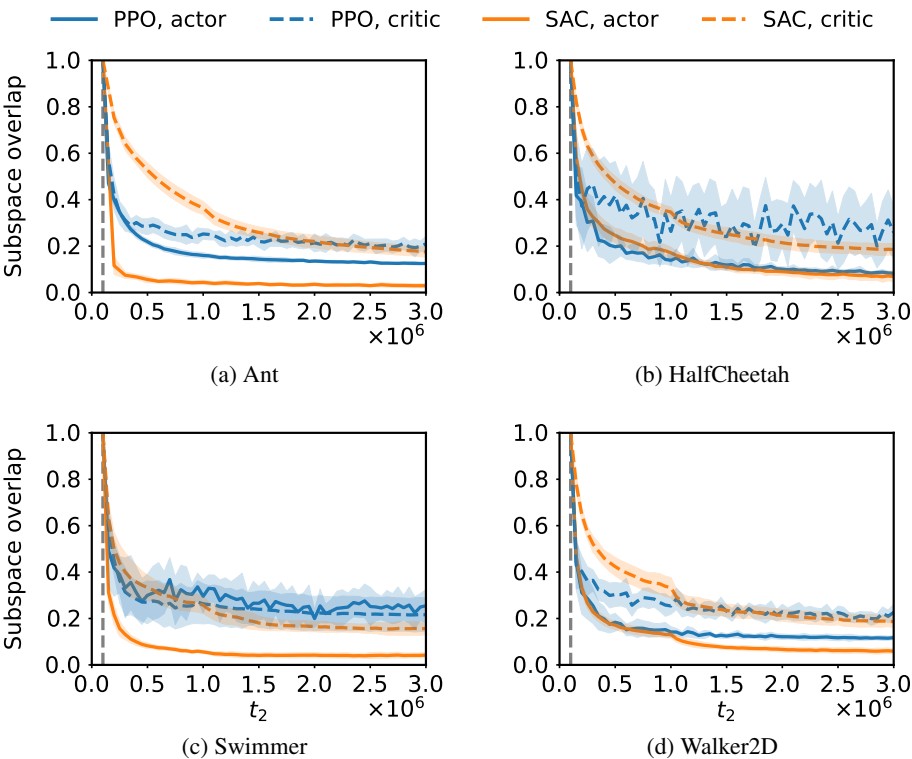

(a) Ant

(b) HalfCheetah

(c) Swimmer

(d) Walker2D

Figure 12: Evolution of the subspace overlap between the early timestep $t_1 = 100{,}000$ (marked by the dashed gray line) and all future timesteps of the training. Results for the actor and critic of PPO and SAC. For SAC, a small drop in the subspace overlap is visible in all plots at around 1.1 million timesteps. This marks the timestep at which the data in the replay buffer is replaced completely by new data, indicating that the data distribution affects the subspace overlap.

