# OpenReview forum: "Identifying Policy Gradient Subspaces"
_ICLR.cc/2024/Conference — ICLR 2024 poster_

### Official Review · Reviewer_862E · 2023-10-24

**Soundness:** 3 good
**Presentation:** 3 good
**Contribution:** 3 good
**Rating:** 6
**Confidence:** 2

**Summary:**

The authors investigate the *existence* of gradient subspaces in policy gradient algorithms. More specifically, they conduct an empirical campaign on two relevant methods (i.e., PPO and SAC) to verify that there exist directions with high curvature spanning a subspace that stays relatively stable throughout the training and that contains the gradient. To this end, the authors propose several numerical setups that verify such an existence. All experiments are conducted both for the actor and the critic gradients. Finally, the authors discuss how the existence of these subspaces could be used in practice to advance the state-of-the-art in policy search algorithms.

**Strengths:**

- The existence of gradient subspaces has gathered consistent attention in the supervised learning community. This work aims to empirically show that such subspaces exist in RL as well. Given the numerous challenges that are introduced in RL, the existence of gradient subspaces is not trivial. In this sense, the work done by the authors aims at filling this gap within the literature, and, therefore, I retain it to be interesting for the community.

- The paper is well-written. The main concepts and idea are easy to understand.

**Weaknesses:**

1. The contribution of the paper is focused on the **empirical existence** of gradient subspaces. Although the authors, in Section 5, discuss how the existence of this sub-space could be leveraged in practice, it remains an open question to provide empirical evidence of these eventual benefits.
2. **Limited experimental campaign**. It has to be noticed that the contributions of this paper are only empirical. As a consequence, I would have expected more experiments to prove the empirical existence of these sub-spaces. Results are limited to 6 domains (i.e., Finger-spin, Ball_in_cup, Ant, HalfCheetah, Pendulum, Walker2D). I invite the authors to expand the set of domains considered.
3. **Clarity and writing (minor)**. I would encourage the authors to introduce and follow a more rigorous and formal definition of the elements that are used in the paper (e.g., subspace and so on). All the discussion is, indeed, rather informal.

**Questions:**

None.

---

> ### Author Response · Authors · 2023-11-20
>
> We thank the reviewer for this precise evaluation and the insightful suggestions. The report helped to improve our paper. We carefully respond to each point below.
>
> > Although the authors, in Section 5, discuss how the existence of this sub-space could be leveraged in practice, it remains an open question to provide empirical evidence of these eventual benefits.
>
> There are potentially many ways to exploit subspaces in the context of PG (parameters space exploration and second order optimization being only two of them). Creating methods that effectively exploit this knowledge is a significant effort that is independent of the analysis. The goal of this paper is to understand gradient subspaces in PG methods and share our findings with the RL community. We hope to spark interest in leveraging this phenomenon for the design of PG methods, similarly to how Gur-Ari (2018) encouraged research on gradient subspaces in supervised learning that resulted in a wealth of practical applications(see the methods mentioned in the related work section). For this reason, we hope for your understanding, that advancing PG algorithms with the findings of this paper is left for future work.
>
>
> > I invite the authors to expand the set of domains considered.
>
> We doubled the number of domains by applying our analysis to six more tasks:  BipedalWalker, Hopper, LunarLanderContinuous, Reacher, and Swimmer from OpenAI Gym (Brockman et al., 2016) and the FetchReach from Gym Robotics (Plappert et al., 2018).
> Please note that the selection of tasks encompasses a wide variety of different characteristics of RL tasks. The tasks range from low-dimensional classical control tasks (Pendulum) to complex high-dimensional locomotion tasks for diverse embodiments (e.g., Ant, Swimmer) and robotics-inspired tasks (FetchReach). Furthermore, the selection includes sparse (Ball_in_cup and FetchReach) and dense reward (e.g., Ant, Walker2d), as well as single-goal (e.g., Ant, Walker2d) and goal-conditioned (FetchReach and Reacher) tasks.
> We expanded the plots in Figure 2 to show results for four (instead of the previous two) tasks. Appendix C displays detailed results for all twelve tasks. Consistently across all tasks, a significant fraction of the gradient lies in the high-curvature subspace. Furthermore, on all tasks, there is a substantial overlap between the subspaces at different timesteps. The consistency of these results across such a diverse set of tasks suggests that stable subspaces are a general phenomenon in PGs rather than being task-specific.
>
> > I would encourage the authors to introduce and follow a more rigorous and formal definition of the elements that are used in the paper (e.g., subspace and so on)
>
> We will add a section to the preliminaries that formally introduces the concept of a subspace and the projections into and out of the subspaces. This section will also state important properties of these matrices. Furthermore, this section will explain the relation between curvature and the Hessian of the objective.
>
> Moreover, we have expanded on the intuition of the criteria in eq. (5) and (7) (eq. (5) and (8) in the revised paper) that we use to evaluate our hypotheses. We added a derivation to Appendix A that shows that the gradient subspace fraction criterion is equivalent to a perhaps more intuitive term (1 - the relative projection error).
>
> We appreciate the insightful comments by the reviewer and hope to have addressed all doubts about our work. We are committed to responding to the remaining questions, if any exist. We would be truly thankful if the reviewer considered raising the score in case no question remains.
>
>
> ### References:
>
> Guy Gur-Ari, Daniel A Roberts, and Ethan Dyer. Gradient descent happens in a tiny subspace. arXiv preprint arXiv:1812.04754, 2018.
>
> Greg Brockman, Vicki Cheung, Ludwig Pettersson, Jonas Schneider, John Schulman, Jie Tang, and
> Wojciech Zaremba. OpenAI Gym. arXiv preprint arXiv:1606.01540, 2016.
>
> Matthias Plappert, Marcin Andrychowicz, Alex Ray, Bob McGrew, Bowen Baker, Glenn Powell, Jonas Schneider, Josh Tobin, Maciek Chociej, Peter Welinder, et al. Multi-goal reinforcement learning: Challenging robotics environments and request for research. arXiv preprint arXiv:1802.09464, 2018.

---

> > ### Comment · Reviewer_862E · 2023-11-20
> > **Ack**
> >
> > I thank the authors for their exhaustive comments. In particular, I appreciated the new set of extensive experiments. I have increased my score accordingly. I have no further questions for the authors.

---

> > > ### Author Response · Authors · 2023-11-21
> > >
> > > We are pleased to read that we addressed the concerns to the reviewer's satisfaction and are thankful for the improved score.

---

### Official Review · Reviewer_CUg2 · 2023-11-01

**Soundness:** 3 good
**Presentation:** 3 good
**Contribution:** 3 good
**Rating:** 6
**Confidence:** 3

**Summary:**

Previous literature demonstrated that the gradient in supervised learning could be dominated by some of several high-curvature subspaces. Inspired by this, the authors investigate whether that is true in policy gradient methods and find that similar phenomena appears in both PPO and SAC by checking the gradient subspace fraction. Moreover, the authors demonstrate that these high-curvature subspaces remain stable across the training process with empirical results.

**Strengths:**

- The paper is clearly presented and easy-to-follow, the motivation and the methodology are clearly described.
- The work has demonstrated that the subspace exists in policy gradient methods as well, similar to what people have discovered in the supervised learning setup. This is done by relatively comprehensive experiments including different approaches to estimate the policy gradient and Hessian matrix and consider both the policy model and the critic model.
- The authors also discuss how to leverage the insights to improve RL algorithms such as subspace-based optimization or parameter-space exploration.

**Weaknesses:**

One of the perspectives to make the paper stronger and more convincing is to show the unique conclusion and domain-specific insight for policy gradient learning, since most of the conclusions actually come from the gradient subspace paper under a supervised learning setup.

**Questions:**

One of the perspectives to make the paper stronger and more convincing is to show the unique conclusion of the policy gradient since this is an extension of the gradient subspace in the supervised learning setup. I could imagine more in-depth discussion in the paper could be helpful. For instances:

- Investigate and explain why the PPO/on-policy method has a much lower gradient fraction in Figure 2, and how much of this is due to the data distribution shift.

 - Have more experiments/configurations in Figure 2 to make the conclusion and discussion more sound, for instance, whether the PPO low gradient fraction is consistent across more RL tasks or not.

 - Have demonstrative experiments to explore one of the potential RL applications (using the insight obtained from this experiment) to make it more convincing that these insights will be helpful for RL training.

---

> ### Author Response · Authors · 2023-11-20
>
> We thank the reviewer for this insightful evaluation, which helped to improve our paper. We carefully answer each point below.
>
> > [S]how the unique conclusion and domain-specific insight for policy gradient learning
>
> The paper already contains insights that are specific to the domain of policy gradient / actor-critic methods.
> Sections 4.2 and 4.3 highlight the differences in the quality of the gradient subspace for the actor and the critic.
> The sections further point out differences between on-policy and off-policy learning.
> The potential application to parameter-space exploration that we discuss in the conclusion is also specific to reinforcement learning.
> Additionally, we are currently running more experiments that directly compare on-policy and off-policy learning by considering an on-policy variant of SAC. These experiment investigate to which extent the differences in the analysis results of SAC and PPO stem from the fact that SAC reuses data and PPO does not.  The results will allow disentangling the effects of on-policy/off-policy learning from the algorithmic details of the two RL algorithms.
>
> > Investigate and explain why the PPO/on-policy method has a much lower gradient fraction in Figure 2, and how much of this is due to the data distribution shift
>
> As mentioned in the response to the first question, we will add experiments that consider an on-policy variant of SAC. Like PPO, the on-policy SAC variant collects a small dataset of on-policy transitions at every update, which is discarded after the update. Since the rest of the algorithm stays the same, these experiments will shed light on whether the differences in the results between PPO and SAC are due to the stronger distribution shift in on-policy learning or due to other details of the algorithms.
>
>
> > Have more experiments/configurations in Figure 2
>
> To increase the confidence that gradient subspaces are a general phenomenon in PG methods, we doubled the number of tasks on which we conducted our analysis by adding the tasks BipedalWalker, Hopper, LunarLanderContinuous, Reacher, and Swimmer from OpenAI Gym (Brockman et al., 2016) and the FetchReach task from Gym Robotics (Plappert et al., 2018). We extended Figure 2 with results for  Ant and LunarLanderContinuous while describing the remaining tasks in Appendix C due to space constraints. Further, we are currently conducting an evaluation of the influence of the hyperparameter configurations on the analysis results, which will be added to the appendix in the revised paper.
>
>
> > Have demonstrative experiments to explore one of the potential RL applications
>
> There are potentially many ways to exploit subspaces in the context of PG (parameter-space exploration and second-order optimization being only two of them). Thus, creating concrete applications of these insights for improving RL requires substantial further investigation and effort. The goal of this paper is to understand gradient subspaces in PG methods and share our findings with the RL community. We hope to spark interest in leveraging this phenomenon for the design of PG methods, similar to how Gur-Ari (2018) encouraged research on gradient subspaces in supervised learning that resulted in a wealth of practical applications (see the methods mentioned in the related work section). For this reason, we hope for understanding that advancing PG algorithms with the findings of this paper is left for future work.
>
> We appreciate the insightful comments by the reviewer and hope to have clarified the aspects in question. We are committed to responding to the remaining questions, if any exist. We would deeply appreciate if the reviewer considered raising the score in case no further doubt remains.
>
> ### References:
> Guy Gur-Ari, Daniel A Roberts, and Ethan Dyer. Gradient descent happens in a tiny subspace. arXiv preprint arXiv:1812.04754, 2018.
>
> Greg Brockman, Vicki Cheung, Ludwig Pettersson, Jonas Schneider, John Schulman, Jie Tang, and
> Wojciech Zaremba. OpenAI Gym. arXiv preprint arXiv:1606.01540, 2016.
>
> Matthias Plappert, Marcin Andrychowicz, Alex Ray, Bob McGrew, Bowen Baker, Glenn Powell, Jonas Schneider, Josh Tobin, Maciek Chociej, Peter Welinder, et al. Multi-goal reinforcement learning: Challenging robotics environments and request for research. arXiv preprint
> arXiv:1802.09464, 2018.

---

> ### Author Response · Authors · 2023-11-22
>
> We hope that our answers and additional experiments address the raised questions to the reviewer's satisfaction. We welcome additional questions if there are any. Should there be no more questions and the reviewer is satisfied, we would be delighted if they considered increasing the score.

---

> ### Comment · Reviewer_CUg2 · 2023-11-22
>
> I am grateful for the authors' recent updates and revisions to the paper, including the updated explanations and the newly conducted experiments.

---

### Official Review · Reviewer_qF4u · 2023-11-01

**Soundness:** 3 good
**Presentation:** 3 good
**Contribution:** 2 fair
**Rating:** 6
**Confidence:** 2

**Summary:**

The papers verifies experimentally the existence of gradient subspace in reinforcement learning in the on-policy algorithm PPO and off-policy algorithm SAC.

**Strengths:**

Current literature on identifying gradient subspace focus on supervised learning, and related work in RL focus on identifying parameter subspace rather than gradient subspace. Therefore, this work is the first to identify gradient subspace in the context of RL and is informative for training RL algorithms. Project codes are provided for reproducibility.

**Weaknesses:**

It is unclear to which extent the contribution of identifying gradient subspace comparing to existing works in RL that focus on identifying parameter subspace (e.g. Gaya et al 2023 in the references) is significant, since both approaches have the same goal of improving training efficiency of policy parameters.

**Questions:**

- Could you elaborate in more detail the motivation of identifying gradient subspace in comparison to parameter subspace, for the goal of guiding parameter-space exploration? This seems to be overlapping with Gaya et al. 2023's claimed benefit of identifying parameter subspace and it is not clear there what would be the benefits of using gradient subspace instead of parameter subspace in that case.
- Is it possible to experimentally verify in a realistic example that the methods of Gaya et al. fail and the methods in current paper succeed?


=============================

Post-rebuttal: I thank the authors for the clarification on their contribution. I have raised my score accordingly.

---

> ### Author Response · Authors · 2023-11-20
>
> We appreciate the reviewer's critical observation and acknowledge that our initial presentation may not have sufficiently highlighted the distinct differences between our work and that of Gaya et al. (2022 and 2023). We're grateful for the opportunity to clarify these distinctions more explicitly.
>
> > It is unclear to which extent the contribution of identifying gradient subspace comparing to existing works in RL that focus on identifying parameter subspace (e.g. Gaya et al 2023 in the references) is significant, since both approaches have the same goal of improving training efficiency of policy parameters.
>
> Both works consider different subspaces and, hence, identifying each subspace can have distinct, maybe even orthogonal, benefits. Gaya et al. generate a new policy through a convex combination of policy parameters that are trained to be distinct (Eq. (6) in Gaya et al. 2022) and maximize the RL objective (Eq. (4) in Gaya et al. 2022). These parameters (referred to as anchor parameters) span the subspace. We, on the other hand, analyze the subspace spanned by the dominant eigenvectors of the Hessian matrix (second derivative of the RL objective wrt. the policy parameters = derivative of the policy gradient).
> Thus, Gaya et al. focus more on generalizing policies, and we appreciate that this line of work also reports benefits of using subspaces. Our work aims to accelerate the vast field of policy gradients methods by using subspaces. In that sense, there might exist multiple ways to exploit subspaces for more efficient policy gradient training. Guiding parameter-space exploration or enabling second-order optimization are just two ways that we propose for future work. In fact, since the works by Gaya et al. from 2022 and 2023 use PPO (line 5 in Figure 2 in Gaya et al. 2022) and SAC (line 8 in Algorithm 1 in Appendix C.1 in Gaya et al. 2023) within their framework, our works could even be combined.
>
> > Could you elaborate in more detail the motivation of identifying gradient subspace in comparison to parameter subspace, for the goal of guiding parameter-space exploration?  This seems to be overlapping with Gaya et al. 2023's claimed benefit of identifying parameter subspace and it is not clear there what would be the benefits of using gradient subspace instead of parameter subspace in that case.
>
> As stated above, we identify and analyze gradient subspaces to make policy gradients more efficient, and the line of work by Gaya et al. proposes to use subspaces of policy parameters to recombine them into new policies. Parameter-space exploration, as formulated in Plappert et al., 2017, can be one way how policy gradient subspace can help advance policy gradient algorithms. We believe that some aspects of the work by Gaya et al. could also be applied for parameter-space exploration. However, it has not been applied to this problem yet.
>
> > Is it possible to experimentally verify in a realistic example that the methods of Gaya et al. fail and the methods in current paper succeed?
>
> Gaya et al. consider policy adaptation (increase the performance of the policy on similar but different MDPs at test time (Gaya et al., 2022)) and continual RL (training agents on sequences of tasks (Gaya et al., 2023)) settings, whereas our work addresses efficiency in single-task learning. So, the objectives are not fully aligned. In fact, the approach by Gaya et al. builds upon learned single-task policies. For that reason, it is not obvious how to create a scenario where both methods can be compared fairly.
>
> We would like to thank the reviewer again for this comment. We hope that this response addresses all of raised doubts about our work. We are committed to answering the remaining ones, if any exist. If there are no further questions, we would really appreciate if the reviewer considered raising the score.
>
> ### References:
>
> Jean-Baptiste Gaya, Laure Soulier, and Ludovic Denoyer. Learning a subspace of policies for online
> adaptation in reinforcement learning. In International Conference of Learning Representations
> (ICLR), 2022.
>
> Jean-Baptiste Gaya, Thang Doan, Lucas Caccia, Laure Soulier, Ludovic Denoyer, and Roberta
> Raileanu. Building a subspace of policies for scalable continual learning. In International Conference of Learning Representations (ICLR), 2023.
>
> Matthias Plappert, Rein Houthooft, Prafulla Dhariwal, Szymon Sidor, Richard Y. Chen, Xi Chen, Tamim Asfour, Pieter Abbeel, and Marcin Andrychowicz. "Parameter space noise for exploration." arXiv preprint arXiv:1706.01905, 2017.

---

> ### Author Response · Authors · 2023-11-21
>
> We are happy that we could clarify the difference between our and Gaya et al.'s work and would like to express our gratitude that the reviewer raised the score.

---

### Official Review · Reviewer_u1A7 · 2023-11-10

**Soundness:** 2 fair
**Presentation:** 4 excellent
**Contribution:** 3 good
**Rating:** 8
**Confidence:** 3

**Summary:**

**EDIT: After the rebuttal, I am raising my score. The authors have promised that they have included additional explanations in the paper (and will add the rest) along with some experiments that they are running, and have modified the abstract to clearly specify the scope of this paper. With these additions, I feel that this paper can be a worthwhile addition to this conference.**

The paper empirically analyzes two popular policy gradient methods (PPO and SAC) on standard RL benchmark tasks and demonstrates (for the first time, to the best of my knowledge) that the actor and critic gradients lie in a tiny subspace (containing about 1% of the original network parameters) and that this subspace changes very slowly. Similar knowledge, as the paper discusses, about the gradients of neural networks in supervised learning have already resulted in various methods to improve training. This paper has the potential to result in similar advances for RL.

I propose a weak accept for this paper because
(0) The findings seem novel and can have a positive impact on PG algorithm development
(1) I find the empirical methods of this paper ad-hoc and not well justified; and
(2) Many empirical details are missing (although, maybe they are all minor details, and I'm over-estimating their importance, especially since the authors released their code);

Note to the authors: I have a limited experience in deep learning, and as such my knowledge of most of the references and the methods employed in this paper is very limited. So if you feel that I don't understand the significance of your analysis, I would appreciate if you could point it out in your rebuttal (and finally in the revised paper). Thanks!

**Strengths:**

**Originality and significance:** The paper (empirically) shows that for the first time that policy gradients (and the critic gradients) of the widely popular deep RL methods (PPO and SAC) lie in a very tiny subspace and this subspace remains somewhat stable during the course of learning. This result is interesting in itself as it gives us additional insights into the deep RL methods. Further, such a result can have significant implications on developing policy gradient methods, as outlined by the authors: Since deep learning methods often have a large number of parameters, this paper's insights can speed up existing RL methods by restricting optimization to a small fraction of those parameters. It can also result in better exploration techniques.

**Quality and clarity:** The paper does a great job of outlining existing research: it's literature survey and references to relevant works is very thorough. The main ideas of the paper themselves are presented in an easy to follow manner, complemented by clear graphs and intuitive explanations. The writing itself is free from any grammatical mistakes (which is uncommon in the papers I have reviewed; so that is nice!).

**Weaknesses:**

# Major issues (these affect my score significantly):

The major weakness of the paper is a lack of rigor and concrete results.

## 1. There is a very weak link between the experimental results and the claims made in the paper: ##

(a) For example, see the following claims of the paper:
- Section 1 (last paragraph): "(i) parameter-space directions with significantly larger curvature exist in PG": --> why can we say that the curvature is significantly larger? Figure 1 is highly qualitative in nature. In fact, Figure 2 is essentially qualitative as well. (And larger than what?)
- Section 1 (last paragraph): "(iii) the subspace is sufficiently stable to be useful for training" --> how do the experiments justify the sufficiency? As far as I understand, Figure 3 only shows that the magnitude of the largest eigenvector projected to the subspace is, say, 0.4 for most part of the learning. What does 0.4 mean? How does that imply sufficient stability for learning based on methods that make updates in this subspace?
- Section 1 (last paragraph): "observe that the value function subspace often exhibits less variability and retains a larger portion of its gradient compared to the PG subspace" --> inconclusive (Figure 2 shows very similar trends for actor and critic)

(b) The results for the paper are specifically for well-tuned PPO and SAC agents on specific RL benchmarks, while the conclusions are drawn for general results. For instance, can we be sure that the subspace would also exist for a random hyperparameter configuration? The existence of subspace is even more important for a wide range of hyperparameters, since we won't know the optimal choice apriori for a random problem. Would the subspace be as restricted (for instance a dimensionality of 0.1% of the total parameter values) for a random SAC agent?

(c) Section 4.3 (paragraph 3): "Similar to the gradient subspace fraction results from Section 4.2, the subspace overlap is more pronounced for the critic than the actor." --> why? Are the scales comparable for the actor and critic graphs in Figure 3? Is this "extra" overlap really useful for designing algorithms? (Maybe it is, but without any explanations or additional experiments, this information is just speculative.)

## 2. There is no rigorous motivation about various metrics used: ##
It seems that the paper adopted the metrics from Gur-Ari et al. (2018) for its experiments. However, these metrics seem somewhat arbitrary, and it is unclear how they relate to the usefulness/existence of the gradient subspace.

(a) Gradient subspace fraction (Eq. 5) seems uninformative. Why showing that the gradient norm in the projected subspace is similar to the original gradient norm helpful? For instance, from the [Johnson-Lindenstrauss lemma](https://en.wikipedia.org/wiki/Johnson%E2%80%93Lindenstrauss_lemma) we know that any random projection matrix will ensure that the projected gradient norm is not too far away from the actual gradient norm. My point being, the metric in Eq. 5 doesn't seem to be useful right away. Maybe one way to make it more informative would be to establish a scale (so compare the norm of the projected gradient to the Hessian eigenvector subspace and the norm of the gradient projected to random subspaces).

(b) Subspace overlap (Eq. 7): This metric is very complicated and I don't clearly see why it helps with showing the stability of the subspace and what that means for restricting training there. Maybe additional references could help with this.

## 3. Missing details about the implementations: ##

I give some examples below:
- Section 4.3 (the equation for S_overlap): what is $k$ in the experiments? Does $k$ change with $t$ in the graph?
- Section 4.3: "where $v_i$ is the ith largest eigenvector at timestep t" --> how is the largest eigenvector determined? Is it by the size of the corresponding eigenvalue?
- Section 4 (paragraph 2): This section is unclear and doesn't provide sufficient details for reproduction (I understand the code does that, but without some details, the paper's results are very difficult to reason about). For instance, SAC usually has multiple Q networks; which ones do the experiments analyze?
- Page 6, Figure 2: I was not able to understand the difference between the three categories: "Estimated/true" gradient, "estimated/true" Hessian, and why does that matter (probably Hessian is used to find the projection matrix $P_k$; if so, please explicitly mention it somewhere)? For instance, why do we care about estimated gradients? I understand that estimated Hessian can be important because we will use it to identify the policy gradient subspace and then use it downstream in algorithms. Some more justification about this could be nice.
- Page 6 (first paragraph): Why was 2%, 0.14%, and 0.07% chosen? These seem highly arbitrary choices. How do the trends change when these numbers are changed?
- Page 5: cutoffs for Equation 6 seem arbitrary.

# Minor issues / suggestions (these do not affect my score as much; please ignore them if you don't agree):
- The abstract seems incomplete/misleading in its present form, and it overclaims the contributions of the paper. For instance, it says "we demonstrate the existence of such gradient subspaces for policy gradient algorithms." While this is not technically wrong to say that, the paper only empirically shows these subspaces, and that too for a very limited class of algorithms, i.e. deep policy gradient methods (in fact just two algorithms PPO and SAC). This could be easily rectified, for instance, by including this line from the introduction: "This paper conducts a comprehensive empirical evaluation of gradient subspaces in the context of PG algorithms, assessing their properties across various simulated RL benchmarks." Further, the line "Our findings reveal promising directions for more efficient reinforcement learning, e.g., through improving parameter-space exploration or enabling second-order optimization." seems to be suggesting that the paper also introduces methods for exploiting these subspaces, whereas that is just suggestion for future research.
- The appendices are not properly referenced anywhere in the paper. At the very least, each section of the appendix should be referenced at the appropriate place in the main paper, explaining what additional details are there. Currently, appendices are a dump of graphs with no accompanying textual explanations (other than the caption).
- Do you think the "LoRA: Low-Rank Adaptation of Large Language Models" (https://arxiv.org/abs/2106.09685) paper could be added to the related works section as well?
- In Section 3.1: can $\gamma$ be 1? Is the setting episodic?
- In Section 3.1 (paragraph 2): The phrase, "expected cumulative reward" could be replaced by "expected (discounted) cumulative reward"
- In Section 3.1 (paragraph 2): "advantage function ... and can also be defined as" --> why say "can also be defined as"? Isn't that the definition of advantage function? Maybe rephrase the sentence..
- Section 3.2, first line: Maybe give a reference for the objective function $J(\theta)$?
- Section 3.2, first line: Also, the definition of the expectation is unclear: in particular, in its current form, the expression $J = \mathbb{E}[ \pi(a_t | s_t) \hat A_t]$ seems to depend on the timestep $t$. That shouldn't be the case..
- Section 3.2, first paragraph: What "estimator of advantage function" is being used? It is not specified.
- Section 3.2, first paragraph (above equation 2): what is the target value for the critic?
- Section 3.3 (above Eq. 4): "exponential of the learned Q-function" --> isn't saying something like a Gibbs distribution or softmax distribution more appropriate?
- Figure 2: since the "true gradient" is just the gradient computed using more state-action pairs, maybe calling it "true" is not accurate?

**Questions:**

I would really appreciate if the authors could clarify the following points (please refer to the Weaknesses section for details):

1(b) Is my understanding correct here? Would these conclusions continue to hold for randomly chosen hyperparameters?

2(a) Why can Eq. 5 helpful? How would the projections on random subspaces look like?

---

> ### Author Response · Authors · 2023-11-20
> **Response to reviewer u1A7 (1/3)**
>
> We thank the reviewer for this detailed and thorough evaluation of our work and the insightful questions. The report improved our paper significantly. We carefully answer each point below.
>
> ## 1.
> (a)
>
> > [W]hy can we say that the curvature is significantly larger?
>
> The curvature is determined by the magnitude of the Hessian eigenvalues. In Figure 1, we show for the actor and critic losses that a handful Hessian eigenvalues exist with a significantly larger magnitude than the rest. Therefore, a small number of parameter-space directions dominate the curvature of the objective.
>
> > And larger than what?
>
> We added “compared to the other parameter-space directions” to make this point clearer.
>
> > "(iii) the subspace is sufficiently stable to be useful for training" --> how do the experiments justify the sufficiency?
>
> Our current experiments do not yet evaluate the sufficiency of the subspace for downstream applications. We changed the formulation to “(iii) the subspace remains relatively stable throughout the RL training.” Although it is yet to be shown that the stability of the subspace throughout the training is useful, we believe this result by itself is already interesting due to the high variance nature of PG algorithms.
>
> > "observe that the value function subspace often exhibits less variability and retains a larger portion of its gradient compared to the PG subspace" --> inconclusive (Figure 2 shows very similar trends for actor and critic)
>
> We added results for more tasks to Figure 2. Furthermore, we fixed a bug in our implementation that changed the analysis results for the Walker2D task (see the general response for details about this bugfix). The updated Figure 2 shows the trend that the critic gradient tends to lie better in the respective subspace more clearly. This effect is particularly strong for PPO, but the average gradient subspace fraction is also slightly higher for the critic in SAC. Please note that this trend is also noticeable in the results for all tasks (see Appendix C) and is not limited to the four tasks displayed in Figure 2.
>
> (b)
>
> This is an important point since hyperparameters in RL generally influence the learning performance significantly. We are currently running an experiment, where we sample hyperparameters in the bounds that we also use for hyperparameter tuning to obtain suboptimal but realistic hyperparameters. We will add the analysis results for these configurations to the revised paper once the experiments are finished.
>
> (c)
> > "Similar to the gradient subspace fraction results from Section 4.2, the subspace overlap is more pronounced for the critic than the actor." --> why? Are the scales comparable for the actor and critic graphs in Figure 3?
>
> Yes, the scales are comparable. We applied the same criterion and used the same subspace dimensions for both the actor and the critic. The criterion is inherently normalized to the range [0, 1] (since we assume the eigenvectors to be norm 1). Furthermore, both networks have the same hidden layer size and, thus, contain roughly the same number of parameters (there is a small difference due to the different input/output dimensions).
>
> The results in Figure 3 and the detailed results in Figures 7 and 8 of Appendix C show that the subspace overlap tends to be larger for the critic than for the actor.
>
> > Is this "extra" overlap really useful for designing algorithms?
>
> The design of algorithms that exploit the subspace phenomenon in the context of PG algorithms can take various forms and, hence, requires substantial experimental validation, which is left for future works. Because there is a significant overlap between the subspaces at different timesteps, the subspace does not need to be determined anew at every step during training. Downstream applications could exploit this fact to increase the computational efficiency of the method. The fact that the subspace overlap is on average higher for the critic than the actor could be exploited for further efficiency gains. The scale of these efficiency gains, however, depends on the specific algorithm.

---

> > ### Author Response · Authors · 2023-11-20
> > **Response to reviewer u1A7 (2/3)**
> >
> > ## 2.
> > (a)
> > > Why showing that the gradient norm in the projected subspace is similar to the original gradient norm helpful? For instance, from the Johnson-Lindenstrauss lemma we know that any random projection matrix will ensure that the projected gradient norm is not too far away from the actual gradient norm.
> >
> > The criterion essentially measures how much of the original gradient is preserved by the projection into the low-dimensional subspace. We regret to have not made it explicit in the original submission that the criterion $\frac{||P_k g||^2}{||g||^2}$ is equivalent to $1 - \frac{||\tilde{g} - g||^2}{||g||^2}$, where $\tilde{g}$ is the gradient first mapped into the subspace and then back to the original dimensionality. The latter criterion measures how well the original gradient can be reconstructed from its subspace representation. This measure exactly represents what matters in our scenario. However, the equivalence above holds only for semi-orthogonal matrices (matrices where the columns are norm 1 and orthogonal to each other). We added this explanation to Section 4.2. of the paper and added the derivation of the equivalence in Appendix A.
> >
> > The random matrices that the Johnson-Lindenstrauss lemma considers are not semi-orthonormal (see e.g. the matrix in  https://www.cs.cornell.edu/courses/cs4786/2020sp/lectures/lecnotes4.pdf). While their rows are approximately orthogonal due to the high dimensionality, they are not norm 1. In essence, this random projection still loses information while mapping to the subspace, it just stretches the vector back to the original length due to its row norm > 1. The equivalence of the criteria above does not hold for the random projection, and, importantly, the relative projection error will be high for such a projection (as it is unlikely that a random subspace contains the gradient by chance).
> >
> > > Maybe one way to make it more informative would be to establish a scale (so compare the norm of the projected gradient to the Hessian eigenvector subspace and the norm of the gradient projected to random subspaces).
> >
> > For random projection matrices, the value of the gradient subspace fraction is in expectation $k/n$, i.e., the ratio of the subspace dimension to the original parameter space dimension (so 0.02 for PPO and 0.0014 for SAC). While we currently do not have a formal proof of this statement, we implemented an empirical sanity check by mapping vectors to random subspaces. The results of this sanity check conformed to our intuition.
> >
> >
> > (b)
> >
> > As for the gradient subspace fraction criterion (eq. (5) (and (6) in the updated paper)), we measure to which extent a vector lies in the subspace. In contrast to the gradient subspace fraction criterion, for the subspace overlap, we do not project the gradient into the subspace, but rather the vectors that span the subspace (the Hessian eigenvectors) of the later timestep $t_2$. Since the subspace is spanned by multiple eigenvectors, we average the criterion over these vectors. The criterion measures how similar the two subspaces are. If the two subspaces are identical (i.e., the subspace did not change over time), the value of the criterion is 1. If there is no overlap between the subspaces, the criterion is 0.
> >
> >
> > ## 3.
> > (a) Throughout the analysis, the subspace dimensionality k remains fixed. As in Section 4.2, we use k=100. We added this clarification to Section 4.3.
> >
> > (b) Yes, we use “i-th largest eigenvector” as a shorthand for ”eigenvector that corresponds to the i-th largest eigenvalue”. We will add this clarification to the preliminaries.
> >
> > (c) The Q-loss of SAC is typically implemented as the mean over the losses for the individual networks, so a single gradient is calculated for the two Q-functions. Hence, we analyze the gradient of both networks.

---

> > > ### Author Response · Authors · 2023-11-20
> > > **Response to reviewer u1A7 (3/3)**
> > >
> > > ## 3 (cont.)
> > > (d)
> > > > I was not able to understand the difference between the three categories: "Estimated/true" gradient, "estimated/true" Hessian
> > >
> > > The difference between the estimated and true gradient / Hessian (note that we renamed these to *mini-batch* and *precise* Hessian in the paper; see answer 4(l)) lies in the number of samples that we use to compute each of the quantities (10⁶ samples vs. a mini-batch). We investigate the “true” quantities to analyze whether the phenomenon occurs in the individual underlying optimization problems that PPO and SAC attempt to solve and the “estimated” quantities since these low-sample approximations are used in practice during training.
> > >
> > > > Why do we care about estimated gradients?
> > >
> > > The “estimated gradients” are mini-batch estimates of the true gradient. RL algorithms like PPO and SAC compute such mini-batch estimates of the gradient for updating the policy. For the analysis results to be useful in practice, it is crucial that the observations also transfer to the mini-batch gradients that are used in practice during training.
> > >
> > > > Hessian is used to find the projection matrix $P_k$; if so, please explicitly mention it somewhere
> > >
> > > We mention at the beginning of Section 4.2 that “the subspaces [are] spanned by the high-curvature directions” and that “$P_k$ denotes the matrix that projects [...] into the subspace”. We further mention in the caption of Figure 2 that the “subspace [is] spanned by the 100 largest Hessian eigenvectors”.
> > >
> > > (e) The subspace dimension is k=100 in the experiments. Different network sizes cause the differing percentages. As mentioned in the paper, we chose k=100 because subspaces of this size can be identified relatively efficiently while  still capturing the gradients to a large degree. In any downstream application, the subspace size would be a hyperparameter, and, depending on the task, smaller values might already suffice. Appendix B (Appendix C in the revised paper) displays analysis results for different subspace sizes.
> > >
> > > (f) We chose these cut-offs because the resulting splits into the different training phases conformed well with our intuition in which phase of the training the agent is at each training step. Since there is no universally accepted notion of when an agent is in an initial training phase / starting to converge, there is not really a way to define these thresholds more rigorously. For this reason, we frankly name this criterion heuristic.
> > >
> > > ## 4.
> > > (a) We modified the abstract to be more precise about our contributions.
> > >
> > > (b) Appendix A (Appendix B in the revised paper) is already referenced in Section 4, and Appendix B (Appendix C in the revised paper) is referenced in Sections 4 and 4.2.
> > >
> > > (c) We will add the LoRA paper to the related work section of the revised paper.
> > >
> > > (d) γ is usually smaller than 1 in practice, but standard definitions often state 0 ≤ γ ≤ 1 (see e.g., chapter 3.3. in (Sutton & Barto, 2018)).
> > >
> > > (e) As suggested, we added *(discounted)* as clarification.
> > >
> > > (f) We changed the formulation to the definition A(s, a) = Q(s, a) - V(s) to make this clearer.
> > >
> > > (g) We added a reference to (Kakade & Langford, 2002) for the objective.
> > >
> > > (h) In this section, we employ the notation from (Schulman et al., 2017) to facilitate readers to check the details of the algorithms in the original paper. The value of the objective depends on the timestep through the states that the agent observes / the actions that it executes at that timestep.
> > >
> > > (i) This part is still about policy gradients in general, so the estimator depends on the concrete algorithm. The advantage estimator for PPO is specified below equation (1).
> > >
> > > (j) We added the definition of the target value (the discounted sum of future episode rewards).
> > >
> > > (k) A Gibbs/Boltzmann distribution would imply the existence of some kind of temperature parameter and is usually defined with a negative exponent. A softmax distribution would suggest that there are multiple (discrete) classes, which is not the case for the (continuous) policy outputs. For the sake of clarity, we stick to the formulation “exponential of the Q-function” from the SAC paper.
> > >
> > > (l) We renamed the “true gradient/Hessian” to “precise gradient/Hessian” and the “estimated gradient/Hessian” to “mini-batch gradient/Hessian” to improve clarity.
> > >
> > > We would like to thank the reviewer for this insightful report and hope to have clarified the aspects in question. We are committed to responding to the remaining questions, if any exist. If there are no further remarks, we would really appreciate if the reviewer considered raising the score.
> > >
> > > ### References:
> > > Richard S Sutton and Andrew G Barto. Reinforcement learning: An introduction. MIT press, 2018.
> > >
> > > Sham Kakade and John Langford. "Approximately optimal approximate reinforcement learning." Proceedings of the Nineteenth International Conference on Machine Learning. 2002.
> > >
> > > John Schulman et al. Proximal policy optimization algorithms. arXiv preprint arXiv:1707.06347, 2017.

---

> > > > ### Comment · Reviewer_u1A7 · 2023-11-22
> > > >
> > > > This was very helpful. Thanks!

---

> > > ### Comment · Reviewer_u1A7 · 2023-11-22
> > >
> > > I appreicate the explanation of point 2(a). This additional explanation (and the results of the sanity check about the random matrix) would be super useful if added to the paper.
> > >
> > > Point 2(b) remains unclear to me.

---

> > ### Comment · Reviewer_u1A7 · 2023-11-22
> >
> > Thanks for the detailed comments. This makes sense. While I cannot check if all these additional experiments / additional explanation has been inclined in the paper, I believe that adding these details will make the paper much more useful and accessible. (It's even worthwhile just to place these in the appendix, if shortage of space is a concern.)

---

> ### Author Response · Authors · 2023-11-22
>
> We hope that our response clarifies all of the reviewer's questions. If any aspect of our paper remains unclear, we encourage the reviewer to pose further questions. Should the reviewer be satisfied with our response and have no more questions, we would be delighted if they considered increasing the score.

---

> > ### Comment · Reviewer_u1A7 · 2023-11-22
> >
> > Your comments were very thorough. I am thankful for that.
> >
> > I have raised the scores. But again, I implore you to include additional details and experiments in  the revised paper.

---

### Author Response · Authors · 2023-11-20
**Response to all reviewers**

We thank the reviewers for their thorough and helpful remarks. We appreciate the time the reviewers invested to provide feedback and hope we have done justice to their comments. We are grateful for the encouraging remarks that suggest our findings “seem novel and can have a positive impact on PG algorithm development” (R1), recognize our work as “the first to identify gradient subspaces in the context of RL” and as “informative for training RL algorithms” (R2), find our paper “clearly presented” and our experiments to be “relatively comprehensive” (R3), and consider our paper “interesting for the community” (R4).
We carefully incorporated the reviewers’ comments and answered them in the respective responses to each reviewer.

The major improvements to the paper are:
* We doubled the number of tasks from six to twelve (as R4 suggested to consider more tasks) and extended Figure 2 to display the results on four instead of two tasks (as R3 suggested). Detailed results for all tasks can be found in Appendix C.
We found that consistently across all tasks, a significant fraction of the gradient lies in the high-curvature subspace. Furthermore, on all tasks, there is a significant overlap between the subspaces at different timesteps, even if these timesteps are far apart.

* We fixed a bug in our implementation that affected the analysis results for PPO on the HalfCheetah and Walker2D tasks. This bug caused a normalization wrapper to not be applied to the environment during the analysis. As a result, during training, the agent operated on the normalized observations and rewards, while the analysis applied the agent loss to the unnormalized observations and rewards. Fixing this bug changed the results on the two tasks and solidified the conclusion that gradient subspaces exist in PPO.

* R1 requested clarifications regarding the intuition of the gradient subspace fraction criterion (eq. (5)). We established an equivalence to a more intuitive criterion (the relative projection error). We derive this equivalence formally in Appendix A.

Furthermore,  we are currently running additional experiments to back up our responses to the comments of the reviewers with data. R1 requested an evaluation of the robustness of the analysis results with respect to the hyperparameters of the agent. R3 asked for experiments that investigate the effect of the shift of the data distribution on the analysis results. To that end, we will add data that compares SAC to an on-policy variant of SAC. This experiment allows comparing the quality of the gradient subspace under larger (in the on-policy case) and smaller (in the off-policy case) distribution shift. We plan to have the results ready by the end of the discussion period.

---

### Meta-Review · Area_Chair_oAdW · 2023-12-11

**Metareview:**

This paper shows that the gradients of PPO and SAC lie in a much smaller space than that of the original gradient and that the subspace changes slowly over time. The result of this work is an understanding that can be used to design more efficient policy gradient algorithms. The reviewers agree that this paper produces useful insights worth sharing with the community. I recommend this paper for acceptance.

**Justification For Why Not Higher Score:**

This work demonstrates a clear insight into optimizing neural network policies and value functions, which is important in the RL community. However, this work confirms a similar result found in supervised learning literature, and thus, it may not be exciting enough to be given a spotlight or oral presentation.

**Justification For Why Not Lower Score:**

This paper provides novel insights into policy optimization and thus deserves acceptance. It can also go on to influence future work in the community. So this paper should have a place at the conference.

---

### Decision · Program_Chairs · 2024-01-16

Accept (poster)